

# Multivariate statistical air mass discrimination for the high-alpine observatory at the Zugspitze mountain, Germany

Armin Sigmund[1,*], Korbinian Freier[2,3], Till Rehm[4], Ludwig Ries[5], Christian Schunk[6,+],
Anette Menzel[6,7], and Christoph K. Thomas[1]

[1]Micrometeorology Group, University of Bayreuth, Bayreuth, Germany
[2]Bavarian Environment Agency, Augsburg, Germany
[3]Research Unit Sustainability and Global Change, Center for Earth System Research and Sustainability (CEN), University of Hamburg, Hamburg, Germany
[4]Environmental Research Station Schneefernerhaus, Zugspitze, Germany
[5]German Environment Agency, GAW Global Observatory, Zugspitze-Hohenpeissenberg, Germany
[6]Ecoclimatology, Technical University of Munich, Freising, Germany
[7]Institute for Advanced Study, Technical University of Munich, Garching, Germany
[*]now at: School of Architecture, Civil and Environmental Engineering, Swiss Federal Institute of Technology, Lausanne, Switzerland
[+]now at: Safety and Radiation Protection, Technical University of Munich, Garching, Germany
**Correspondence:** A. Sigmund (armin.sigmund@epfl.ch)

**Abstract.** To assist atmospheric monitoring at high-alpine sites, a statistical approach for distinguishing between the dominant air masses was developed. This approach was based on a principal component analysis using five gas-phase and two meteorological variables. The analysis focused on the site Schneefernerhaus at Mt. Zugspitze, Germany. The investigated year was divided into 2-month periods, for which the analysis was repeated. Using the 33.3 % and 66.6 % percentiles of the first two
5 principal components, nine air mass regimes were defined. These regimes were interpreted with respect to vertical transport and assigned to the air mass classes ML (recent contact with the mixing layer), UFT/SIN (undisturbed free troposphere or stratospheric intrusion), and HYBRID (influences of both the mixing layer and the free troposphere or ambiguous). 78 % of the investigated year were classifiable. ML accounted for 31 % of the cases with similar frequencies in all seasons. UFT/SIN comprised 14 % of the cases but were not found from April to July. HYBRID (55 %) mostly exhibited intermediate charac-
10 teristics, whereby 17 % of HYBRID suggested an influence of the marine boundary layer or the lower free troposphere. The statistical approach was compared to a mechanistic approach using the ceilometer-based mixing layer height from a nearby valley site and a detection scheme for thermally induced mountain winds. Only 25 % of the cases were classifiable with the mechanistic approach. Both approaches agreed well, except in the rare cases of thermally induced uplift. The statistical approach is a promising step towards a real-time discrimination of air masses. Future work is necessary to assess the uncertainty
15 arising from the standardization of real-time data.



# 1   Introduction

High-alpine observatories such as the Environmental Research Station Schneefernerhaus (UFS) at Mt. Zugspitze, Germany, play an important role in studying changing concentrations of atmospheric constituents such as greenhouse gases, aerosols, and persistent organic pollutants (POPs), which have critical impacts on the climate, environmental integrity, or human health (McClure et al., 2016; Kirchner et al., 2016). In particular, high-alpine observatories frequently offer the opportunity to sample

well-mixed air masses of the free troposphere. These air masses are representative for large spatial areas and thus suitable for the determination of large-scale and global trends (Yuan et al., 2019). Therefore, the observational network of the Global Atmosphere Watch Program of the World Meteorological Organization includes many high-alpine observatories. At times, however, high-alpine sites can be affected by local anthropogenic emissions on the mountain or regional emissions if air masses of the mixing layer (ML) are lifted by processes such as synoptic lifting (e.g. at fronts), thermally induced anabatic

winds, and foehn flows (Zellweger et al., 2003). In relatively rare cases, the air masses originate from the stratosphere, e.g. on approximately 6 % of the days at Mt. Zugspitze (Stohl et al., 2000).

The terms ML and boundary layer can be defined synonymously as the sum of all atmospheric layers that exchange air with the surface during one diurnal cycle (Seibert et al., 2000; Reuten et al., 2007). This definition was proposed to explicitly include the residual layer above a stably stratified nocturnal boundary layer and elevated aerosol layers that can result from thermally

driven upslope flows (Reuten et al., 2007; Gohm et al., 2009). For instance, on typical fair weather days in summer, Henne et al. (2004) observed a two-layer structure of the ML in deep Alpine valleys where upslope flows lifted air from a polluted lower layer to a moderately polluted injection layer that reached well above the crest height.

Basically, there are two different approaches for air mass discrimination at high-alpine sites. Mechanistic approaches investigate directly atmospheric transport processes using meteorological measurements or trajectory models, whereas statistical

approaches infer the influences of the air layers from the air mass composition at the site and, possibly, meteorological auxiliary data.

Zellweger et al. (2003) used a mixed mechanistic and statistical approach at the high-alpine site Jungfraujoch, Switzerland. They identified three uplift processes, namely foehn events, synoptical lifting, and thermally induced uplift, and attributed the other cases to the undisturbed free troposphere. Foehn events and synoptic lifting were detected with mechanistic approaches

using standard meteorological measurements and the height of back-trajectories, respectively. Thermally induced uplift was determined with a statistical approach based on the diurnal variation of the sum of oxidized nitrogen species ($NO_y$) or, in the case of data gaps, the aerosol surface area concentration or specific humidity.

Other studies used a ground-based lidar or ceilometer that was installed near a high-alpine site at a lower altitude to determine the mixing layer height (MLH) from the vertical aerosol backscatter profile (Gallagher et al., 2012; Ketterer et al., 2014). This

method is considered as a mechanistic approach because the MLH is a meteorological quantity. Recently, Poltera et al. (2017) used a tilted configuration of a ceilometer and demonstrated that the Jungfraujoch was rarely embedded in the local convective boundary layer but much more frequently in an above lying injection layer with slightly higher aerosol concentrations compared to the free troposphere.



Some statistical approaches defined a threshold for the concentration of a surface-emitted atmospheric constituent or a ratio of constituents. At the Mt. Bachelor Observatory, USA, a seasonal or monthly threshold for the water vapor mixing ratio was used to distinguish between free-tropospheric and ML influenced air masses (Ambrose et al., 2011; Zhang and Jaffe, 2017). The threshold was chosen such that the water vapor mixing ratios below this threshold had the same seasonal or monthly mean as the data from National Weather Service soundings that were launched at a lower elevated site.

For the Jungfraujoch, Herrmann et al. (2015) compared a mechanistic approach, which was based on back-trajectories and an inventory of carbon monoxide (CO) emissions, with two simple statistical approaches that used a constant threshold of the radon-222 concentration and the ratio of CO to $NO_y$. The $CO/NO_y$ threshold appeared to achieve the best distinction between free-tropospheric and ML influenced air masses but the authors noted that a single threshold cannot account for varying degrees of ML influence (Herrmann et al., 2015).

Other statistical approaches aimed at selecting baseline (also called background) concentrations of a trace gas at different kinds of remote sites and were based on outlier removal techniques (e.g. Ruckstuhl et al., 2012) or the fact that well-mixed air masses result in a small temporal variability of the trace gas mixing ratio (e.g. Yuan et al., 2018). Baseline concentrations refer to a given species in a well-mixed air mass with a minimal influence of anthropogenic impurities of relatively short lifetime (Calvert, 1990; Yuan et al., 2018), which is associated with the free troposphere at high-alpine sites.

In contrast to mechanistic approaches, statistical air mass classifications are not able to distinguish between different uplift processes but require only local data. So far, however, statistical approaches were only based on a single constituent or a ratio of constituents, although several atmospheric constituents are typically monitored at high-alpine observatories.

This study proposes a novel statistical approach based on a principal component analysis (PCA) using seven chemical and meteorological variables. This approach is intended for a later use in real-time operational mode to enable an automated
sampling of ambient air with respect to different air masses using a multi-channel sampling system for monitoring of POPs (Kirchner et al., 2016). The objectives were to (i) develop a statistical classification scheme for the site UFS at Mt. Zugspitze and to (ii) validate this approach, as far as possible, with a mechanistic approach based on ceilometer and standard meteorological measurements.

## 2   Methods

### 2.1   Measurement sites

The UFS (47°25'00" N, 10°58'47" E, 2650 m a.s.l.) is located on a steep south-facing slope, approximately 300 m below the summit of Mt. Zugspitze (2962 m a.s.l.), which is the highest mountain in the German Alps and represents the first real barrier for northwesterly advection from the Alpine foreland. At the UFS, westerly and easterly wind directions dominate due to the local topography (Risius et al., 2015). Because of trace gas and aerosol measurements, the UFS reduced its emissions of
these substances to a minimum. Nevertheless, the measurements can be influenced by local emissions from the direct surrounding, which is a highly frequented tourist area all year round. Local emission sources include nearby cable car stations at the



Zugspitze summit (ZSG) and the Zugspitzplatt (ZPLT) which is a gently sloping plateau below the UFS (Fig. 1). Additionally, a skiing area is situated at the ZPLT. The large metropolitan area of Munich is approximately 90 km north of the study site.

Beside the UFS, seven weather stations at different altitudes at a maximum horizontal distance of 11 km from the UFS were available and included in this study (Fig. 1). One of these stations is located at ZSG. Another weather station is located on the ZPLT which is surrounded by mountain ridges except towards the east where the plateau leads to the narrow and deep valley Reintal. The weather stations Schachen (1830 m a.s.l.), Kreuzalm (1600 m a.s.l.), Felsenkanzel (1250 m a.s.l.), and Brandwiese (900 m a.s.l.) are part of the project KLIMAGRAD (Schuster et al., 2014). The site Schachen is situated on a plateau above the middle Reintal in close vicinity of some trees. Kreuzalm is situated on a meadow on a mountain saddle. Brandwiese is located on a meadow surrounded by forest, where a tributary valley reaches the Reintal from the west. At Felsenkanzel, the measurements are made on a steep south-facing slope northwest of the town of Garmisch-Partenkirchen (GAP). The site GAP (720 m a.s.l) is located in the western periphery of the town where a broad west-east oriented valley turns to northeast towards the alpine foreland (Fig. 1).

## 2.2 Instrumentation and data set

The data set that was analyzed in this study spans a period of 1 year from 22 August 2013 to 21 August 2014, which was selected for high data availability. Table 1 gives an overview of the chemical measurements, associated instruments, measurement principles, and research institutions that provided the data. Most of the atmospheric constituents are measured at the UFS, including carbon mono- (CO) and dioxide ($CO_2$), methane ($CH_4$), ozone ($O_3$), the sum of oxidized nitrogen species ($NO_y$), nitrogen oxides ($NO_x = NO + NO_2$ with NO and $NO_2$ being nitrogen mono- and dioxide, respectively), formaldehyde (HCHO), the ambient particle number size distribution ($dN \, (dlog \, d_p)^{-1}$) for particle diameters ($d_p$) between 10 nm and 600 nm, the mass concentration of particulate matter with $d_p < 10$ μm ($PM_{10}$), and equivalent black carbon (eBC). The radioisotopes beryllium-7 ([7]Be) and radon-222 ([222]Rn) are sampled at ZSG and the mountain ridge (2825 m a.s.l.) directly above the UFS, respectively. At the remaining sites, only meteorological parameters are recorded.

While most of the chemical data were available at 1 min intervals, HCHO and the particle size distribution were provided at 10 min intervals and [222]Rn and [7]Be were available at 2 h and 12 h intervals, respectively (Table 1). The HCHO data contained a large data gap between 15 December 2013 and 15 July 2014 and the [222]Rn data was only available since 1 January 2014. The [222]Rn data were downloaded from the World Data Centre for Greenhouse Gases (WDCGG, 2019). Because [7]Be attaches to aerosol particles, it is measured by collecting the carrier aerosol with a glass fiber filter. While NO can be measured directly, $NO_y$ and $NO_2$ have to be converted to NO before the detection. This conversion of $NO_y$ and $NO_2$ is performed with photolysis and gold/CO converters, respectively.

The standard meteorological measurements included air temperature ($T$), relative humidity (rH), global radiation ($R_g$), and horizontal wind velocity and direction at all sites. Air pressure ($p$) was available at the sites UFS, ZSG, and GAP. Additionally, year-round precipitation measurements with an electronic weighing system and a windbreak ring (Sommer Messtechnik, Koblach, Austria) at the site ZPLT were used. At the ZPLT and the four KLIMAGRAD sites, the wind data is measured by propeller anemometers (Wind Monitor 05103, Young, Traverse City, USA) while at GAP and ZSG, cup anemometers and wind



vanes (SK-565 and SK-566, respectively, Thies Clima, Göttingen, Germany) are used. At the UFS, an ultrasonic anemometer (model 2D, Thies Clima, Göttingen, Germany) is used.

At GAP, a ceilometer (CHM 15k, Lufft, Fellbach, Germany) provides up to three aerosol layer heights (ALHs) with a vertical resolution of 15 m at 15 s intervals using a wavelet algorithm, which detects strong gradients in the range corrected attenuated backscatter profile. In the absence of clouds, the signal-to-noise ratio of the ceilometer is typically $> 1$ up to a height of 4 to 5 km a.g.l. in the daytime and up to greater heights at night (Heese et al., 2010).

## 2.3 Data post-processing and quality control

All time stamps were converted to local standard time. The data were aggregated on a 30 min basis while requiring a data availability of $\geq 66\ \%$ in each interval unless stated differently. The $^{222}$Rn and $^{7}$Be concentrations were interpolated using nearest neighbor interpolation to match them with the 30 min intervals used for the analysis.

### 2.3.1 Atmospheric constituents

The chemical measurements were quality controlled by the research institutions that provided the data. Invalid data, which resulted from calibration, instrument repair, or power failure, were discarded.

Due to measurement uncertainties, $NO_y$ occasionally exhibited a lower mixing ratio than $NO_x$. If $NO_y$ was lower than 75 % of $NO_x$ and $NO_x$ was at least 0.03 ppb, both quantities were treated as artifacts and were discarded. If $NO_y$ remained still lower than $NO_x$ after averaging on a 30 min basis, both quantities were assumed to be equal and were replaced by the mean of $NO_x$ and $NO_y$. Negative eBC concentrations were treated in a similar way. Values below $-0.05\ \mu g\,m^{-3}$ were discarded. If eBC remained still negative after averaging on a 30 min basis, the concentration was set zero.

The processing of the particle number size distributions is described in Birmili et al. (2016) and includes, among others, a multiple charge inversion and corrections for particle losses. Following Herrmann et al. (2015), the number concentration of accumulation mode particles ($N_{90}$) was used as an indicator for ML air masses and approximated by including particle diameters between 90 nm and 600 nm .

Data of most trace gases and $PM_{10}$ had been flagged manually with respect to locally polluted air masses. These air masses are a special kind of ML air masses because they do not result from uplift processes but from human activities on the mountain. Local emissions were evident from short but pronounced peaks in trace gas and aerosol concentrations and were most frequently observed for $NO_x$ and $NO_y$. The flag for local $NO_x$ emissions was approximately reproduced by selecting data with a 30 min $NO_x$ standard deviation ($\sigma_{NO_x}$) of $> 0.4$ ppb (not shown). According to this criterion, local pollution events affected 9 % of the $NO_x$ data but half of these events lasted no longer than one time interval, i.e. 30 min. Consequently, local pollution events are generally so short that they cannot be predicted on a 30 min basis in real-time applications. Therefore and because the classification scheme was developed for a later use in real-time operational mode, the air mass classification does not account for local pollution events.





### 2.3.2 Standard meteorological data

Possible time offsets between the sites were determined and corrected by maximizing the cross-correlation coefficient between global radiation at the UFS and the other sites on 23 September 2013 and 20 March 2014, two days with clear-sky conditions. The cross-correlation was evaluated in a time window where morning and evening hours were excluded because of shadowing effects depending on the site. Using data at 10 min intervals, the time offsets ranged between $-20$ min and $+10$ min.

The wind data at ZSG and GAP were discarded for wind velocities below the starting threshold of the wind vane of $0.3\,\mathrm{m\,s^{-1}}$ for a displacement of $90°$ (Löffler, 2012). For the wind vane of the propeller anemometers at the ZPLT and the KLIMAGRAD sites, the starting threshold was $1.1\,\mathrm{m\,s^{-1}}$ (Young, 2018). To achieve a trade-off between a high data quality and a high data availability, the wind data of these anemometers were discarded for wind velocities of $< 0.7\,\mathrm{m\,s^{-1}}$.

     The standard meteorological data were aggregated on a 30 min basis using the vector mean for wind direction, the sum for

precipitation, and the arithmetic mean for the other variables. Specific humidity $q$ $(\mathrm{g\,kg^{-1}})$ and virtual potential temperature $\theta_v$ (K) were calculated for each site as follows (Foken, 2008):

$$q = 0.622\,\frac{e}{p - 0.378\,e} \tag{1}$$

$$\theta_v = (1 + 0.608\,q)\,T\left(\frac{p_0}{p}\right)^{R_L/c_p} \tag{2}$$

where $e$ (hPa) is vapor pressure, $p$ (hPa) is air pressure, $p_0$ is $1000$ hPa, $T$ (K) is air temperature, $R_L$ $(\mathrm{J\,kg^{-1}\,K^{-1}})$ is the gas constant of dry air, and $c_p$ $(\mathrm{J\,kg^{-1}\,K^{-1}})$ is specific heat capacity of dry air. The vapor pressure was calculated from relative humidity (rH) and $T$ using the Clausius Clapeyron equation. Because $p$ was only measured at the sites ZSG, UFS, and GAP, it was calculated for the other sites using the hypsometric equation:

$$p = p_{ref}\,exp^{-\frac{g_0\,(Z - Z_{ref})}{R_L\,T_v}} \tag{3}$$

where $p$ (hPa) and $p_{ref}$ (hPa) are the air pressures at the site of interest and a reference site, respectively, $Z$ (m) and $Z_{ref}$ (m) are the geopotential heights that were approximated with the altitudes of the respective sites, $g_0$ is $9.8\,\mathrm{m\,s^{-2}}$, $R_L$ $(\mathrm{J\,kg^{-1}\,K^{-1}})$ is the gas constant of dry air, and $T_v$ (K) is the mean virtual temperature for the air layer between the two sites under consideration. $p$ was computed in two steps. First, $p$ was approximated using the dry-bulb temperature instead of $T_v$. Second, $T_v$ was calculated with the approximated $p$ and used to recalculate $p$. The UFS was used as the reference site to calculate $p$ at the

next lower site (ZPLT), which was afterwards used as the reference site to calculate $p$ at the next lower site (Schachen). This procedure was continued until Kreuzalm. Similarly, GAP was used as the reference site to determine $p$ at the next higher site and this procedure was continued until Schachen. At Kreuzalm and Schachen, the two $p$ estimates based on the next lower and next higher sites were averaged.



In order characterize the static stability of the valley atmosphere with a single quantity, the range of the pseudo-vertical profile of virtual potential temperature ($\Delta\theta_v$ (K)) was defined as follows, considering all stations except Schachen:

$$\Delta\theta_v = \begin{cases} \theta_v^{\min} - \theta_v^{\max}, & \text{if } z_{\theta_v^{\min}} \leq z_{\theta_v^{\max}} \\ \theta_v^{\max} - \theta_v^{\min}, & \text{if } z_{\theta_v^{\min}} > z_{\theta_v^{\max}} \end{cases} \tag{4}$$

where $\theta_v^{\min}$ (K) and $\theta_v^{\max}$ (K) are minimum and maximum virtual potential temperature of the pseudo-vertical profile, respectively, and $z_{\theta_v^{\min}}$ (m) and $z_{\theta_v^{\max}}$ (m) are the associated altitudes. Schachen was excluded from this calculation because $\theta_v$ was

particularly high at Schachen if the global radiation was high and the wind velocity was low, which suggested that the naturally aspirated thermometer was affected by radiative errors (not shown). $\theta_v^{\min}$ was mostly found at the low-elevation sites GAP and Brandwiese, which applied to 57 % and 37 % of the cases, respectively. $\theta_v^{\max}$ occurred mostly at the high-elevation sites ZSG, UFS, or ZPLT, which applied to 68 %, 18 %, and 13 % of the cases, respectively.

### 2.3.3 Ceilometer data

To determine the MLH at GAP from the ALH retrievals of the ceilometer, several post-processing steps were necessary (Fig. 2). ALHs below 600 m a.g.l., corresponding to 1320 m a.s.l., were regarded as artifacts and were discarded because of high measurement uncertainties in this part of the backscatter profile resulting from an incomplete overlap of the laser beam and the field of view of the ceilometer (Flentje et al., 2010). The backscatter signal only reflects aerosol concentrations below clouds. Thus, ALHs that were greater or equal to the cloud base height, which was measured by the same instrument, were discarded.

For this reason, it was not possible to determine the correct MLH if clouds were present at the top of or within the ML. These first two post-processing steps strongly reduced the data availability from 89 % to 45 % of the time.

If two or three ALHs were found at the same time, the following procedure was applied. For each ALH, the number of ALHs in its vicinity, i.e. within centered intervals of 300 m and 30 min in the vertical and temporal dimensions, respectively, was determined. The ALH with the maximum number of ALHs in its vicinity was attributed to the MLH while the other ALHs were

discarded. If each of the ALHs had less than three ALHs in its vicinity, a missing value was registered. The MLH attribution, as described above, was based on the idea that the MLH varies only gradually so that similar MLHs are expected shortly before and after the time under consideration.

From the resulting time series at 15 s intervals, outliers were removed iteratively as follows. From each pair of subsequent data points with an absolute rate of change exceeding $240 \text{ m min}^{-1}$, one data point was removed so that the standard deviation

of the MLH in the preceeding 15 min and the following 15 min was most strongly reduced. This procedure was repeated until the absolute rate of change was maximum $240 \text{ m min}^{-1}$ corresponding to four vertical averaging intervals ($4 \cdot 15$ m) per time interval (15 s).

The MLHs were aggregated using the 30 min median while requiring a data availability of $\geq 50$ % and a standard deviation of $\leq 170$ m corresponding to the 90 % percentile of all 30 min standard deviations. From the aggregated time series, outliers

were removed in the same way as described above but with a maximum allowed absolute rate of change of $400 \text{ m h}^{-1}$. Finally, the MLH was available for 34 % of the time.

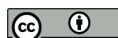



## 2.4 Statistical classification approach

### 2.4.1 Definition of the air mass classes

The statistical analysis aimed at distinguishing the following four classes of air masses: (a) ML: Air masses with a recent contact with the ML, characterized by increased concentrations of surface-emitted atmospheric constituents suggesting a recent uplift with a dominating influence of regional or local sources; (b) UFT/SIN: Air masses of the undisturbed free troposphere or

stratospheric intrusions, characterized by very low concentrations of surface-emitted constituents suggesting recent subsidence or horizontal advection; and (c) HYBRID: Air masses that are influenced by both the ML and the free troposphere or that exhibit ambiguous characteristics. The class HYBRID was anticipated, for example, if the air mass had been exported from the ML to the free troposphere at a significant horizontal distance from the UFS and had been mixed with the free troposphere on the further trajectory, resulting in intermediate concentrations of surface-emitted constituents.

Stratospheric intrusions are characterized by high $^7$Be and $O_3$ concentrations and a low humidity and have been observed at Mt. Zugpitze on approximately 5 % of the days (Stohl et al., 2000). Because of this low frequency and the coarse temporal resolution of $^7$Be of 12 h, stratospheric intrusions were not classified individually but attributed to the same class as air masses of the undisturbed free troposphere, which exhibit similar characteristics.

### 2.4.2 Principal component analysis

A PCA allows for reducing the number of dimensions of a data set while maintaining as much variance as possible. Principal components (PCs) are uncorrelated standardized linear combinations of the original variables (Mardia et al., 1979). If the variance of the original variables is mainly caused by shifts between the air mass classes, the first few PCs will be suitable indicators for air mass discrimination. The PCA was performed separately for 2-month periods to (i) account for seasonally changing relationships between the variables and to (ii) largely eliminate the influence of the seasonal variability of atmospheric

constituents because the seasonality reflects not only the frequency of ML influenced air masses but also other factors such as the source and sink strength (e.g. residential heating in winter, photosynthetic $CO_2$ removal in summer), chemical reactivity, and deposition. The original variables were standardized by subtracting the arithmetic mean and dividing by the standard deviation of the respective 2-month period in order to avoid biased results due to different units and ranges of the data.

Only the most suitable variables were used as input variables of the PCA whereas the other variables were used for validation

purposes. Aerosol measurements were excluded from the PCA because low aerosol concentrations do not necessarily indicate UFT/SIN air masses but can also result from wet deposition during the uplift of ML air masses. $^{222}$Rn and HCHO were only used for validation purposes due to low data availability. $NO_y$ and $NO_x$ were excluded from the PCA because their variability was particularly strongly influenced by local emissions, which are not indicative of uplift processes. $O_3$ was only used as PCA input variable in the winter half year because high $O_3$ mixing ratios are not only caused by a subsidence of UFT/SIN

air masses but can also result from photochemical $O_3$ production in ML air masses, which typically contain high precursor concentrations. In the winter half year, photochemical $O_3$ production was assumed to play a minor role at high-alpine sites because of the generally low solar irradiance and the weak thermally induced uplift. This argument is supported by trajectory



residence time statistics of Kaiser et al. (2007), which demonstrated that in winter, the $O_3$ mixing ratio at European high-alpine sites was generally lower if the air mass originated from lower altitudes, whereas in summer, this was often not the case. Among the meteorological variables, the air pressure at GAP and $\Delta\theta_v$ were considered most suitable because they are physically linked to uplift (low pressure, low static stability) and subsidence processes (high pressure, high static stability).

The 2-month periods were defined as December to January, February to March, ..., so that the winter and summer half years
included the months with the lowest and highest solar forcing, respectively. The PCs were computed from the following five gas-phase variables and two meteorological variables, where $O_3$ was only included in the 2-month periods of the winter half year (October to March).

$$\text{PC}_i = a_{1i}\,\widehat{[\text{CO}]} + a_{2i}\,\widehat{[\text{CH}_4]} + a_{3i}\,\widehat{[\text{CO}_2]} + a_{4i}\,\widehat{[\text{O}_3]} + a_{5i}\,\widehat{q} + a_{6i}\,\widehat{\Delta\theta_v} + a_{7i}\,\widehat{p_{\text{GAP}}} \tag{5}$$

Here, $\text{PC}_i$ represents the scores of the i-th PC; $\widehat{[\text{CO}]}, \widehat{[\text{CH}_4]}, \widehat{[\text{CO}_2]}$, and $\widehat{[\text{O}_3]}$ are the standardized mixing ratios of the respective
trace gases at the UFS; $\widehat{q}$ is the standardized specific humidity at the UFS; $\widehat{\Delta\theta_v}$ is the standardized range of the pseudo-vertical profile of virtual potential temperature; $\widehat{p_{\text{GAP}}}$ is the standardized air pressure at GAP; and $a_{1i}, \ldots, a_{7i}$ are the loadings of the i-th PC. All quantities in Eq. 5 are dimensionless due to standardization. The loadings represent standardized eigenvectors of the correlation matrix of the original variables while the associated eigenvalues correspond to the variances of the PC scores (Mardia et al., 1979).

**2.4.3   Isolating regimes and classes of air masses**

In each 2-month period, the air masses were divided into nine regimes using the 33.3 % and 66.6 % percentiles of the first two PCs as thresholds (Fig. 3, 8b). These regimes were interpreted with respect to vertical transport and attributed to the three air mass classes ML, UFT/SIN, and HYBRID by comparing summary statistics of the PCA input variables between the regimes. The attribution was validated using summary statistics of the remaining measurements. Long-range transport of
mineral dust (LRMD) was regarded as a subclass of HYBRID and was identified with the following three criteria: (i) a high $PM_{10}$ concentration of $\geq 13\ \mu\text{g}\,\text{m}^{-3}$, (ii) a relatively low 30 min standard deviation of the $PM_{10}$ of $\leq 0.2\ PM_{10}$ to avoid the attribution of local pollution events, and (iii) an air mass regime that does not suggest a current uplift of ML air masses. The $PM_{10}$ threshold of $13\ \mu\text{g}\,\text{m}^{-3}$ was motivated by a local minimum in the $PM_{10}$ histogram for non-ML air masses in June–July (not shown) and the plausibility was checked by visual inspection of the whole $PM_{10}$ time series. Ambiguous air mass regimes,
which carried the fingerprint of the marine boundary layer or the lower free troposphere (among others, very low CO and $CH_4$ mixing ratios but high $q$ and low $O_3$ mixing ratios) were attributed to MBL/UFT, another subclass of HYBRID, unless the criteria for LRMD were fulfilled (Fig. 3).

**2.5   Mechanistic classification approach**

In search of criteria for thermally induced anabatic and katabatic winds that influence the UFS, the wind directions of all sites
were investigated. The wind direction at the UFS was not useful because upslope and downslope flows were not clearly visible but appeared to be superimposed by synoptically driven winds due to the relatively exposed location of the UFS (not shown).



Only at GAP, Felsenkanzel, and ZPLT, the wind directions exhibited pronounced diurnal patterns that indicated thermally induced mountain winds, especially in summer (Fig. 4). However, the wind data from Felsenkanzel were not used because of problems with shifts of the sensor orientation. Upvalley and downvalley winds were characterized by northeasterly and southwesterly wind direction sectors at GAP, respectively, and by easterly and westsouthwesterly wind direction sectors at the ZPLT, respectively (Fig. 4). The wind patterns at the ZPLT were consistent with the study of Gantner et al. (2003) who

observed a thermal circulation above the same plateau with an easterly inflow of a few hundred meters depth in the daytime and a westerly outflow at night under fair weather conditions in summer. Modeling suggested that the inflow ascended the narrow and deep valley Reintal before reaching the ZPLT.

The wind direction sectors at the ZPLT and GAP were used as a criterion for anabatic and katabatic winds (Table 2). Additionally, anabatic and katabatic winds were restricted to cases with a weak static stability ($\Delta\theta_v > -8$ K) and a strong

static stability ($\Delta\theta_v \leq -8$ K), respectively, which was justified by the intersect between the frequency distributions of $\Delta\theta_v$ for the wind direction sectors described above (not shown). For katabatic winds, a low wind velocity of $< 3$ m s$^{-1}$ was required at the ZPLT and GAP to exclude cases with a strong synoptic forcing. Additionally, both anabatic and katabatic winds were required to persist for at least 1 h, i.e. two time intervals (Table 2).

The detection of these winds and the ceilometer-based MLH at GAP were combined by a mechanistic approach that primarily

accounted for thermally induced vertical transport and distinguished between the following three conditions: (a) Anabatic winds occur or the UFS is below MLH$_\text{GAP}$, which suggests ML air masses; (b) Katabatic winds occur and the UFS is above MLH$_\text{GAP}$, which suggests UFT/SIN or HYBRID air masses; (c) The UFS is above MLH$_\text{GAP}$ but the winds are not thermally induced, which also suggests UFT/SIN or HYBRID air masses under the assumption of identical MLHs at the UFS and GAP. This approach was limited by the low availability of the MLH of 34 % but still allowed for a partial verification of the statistical

approach.

## 3 Results and discussion

### 3.1 Principal components and their seasonal dependence

The PCA converts several input variables into the same number of PCs while the variance and thus the importance of the PCs is highest for the first few PCs. In the winter half year, the percentage of explained variance of the PCs decreased strongly

from PC1 to PC2 and more slightly towards higher-order PCs. In the summer half year, the explained variance decreased approximately linearly with increasing PC number (Fig. 5). The first two PCs explained a total of 60 % to 72 % and 53 % to 58 % of the variance in the winter half year and summer half year, respectively. Hence, most of the variance was maintained when reducing the number of considered PCs from six or seven to two, especially in the winter half year.

The loadings of the first two PCs were similar among the 2-month periods of the winter half year but more variable within the

summer half year (Fig. 6). Although the loadings differed between the 2-month periods, PC1 was always a meaningful indicator for air mass discrimination. PC2, as well as higher-order PCs, did not always allow for an unambiguous interpretation.



In the winter half year, low scores of PC1 primarily reflected high CO, $CO_2$, and $CH_4$ mixing ratios, a rather high $\Delta\theta_v$ (i.e. rather low static stability), and a rather low air pressure at GAP ($p_{GAP}$), which suggested an uplift of ML air masses, while high scores of PC1 reflected the opposite characteristics suggesting a subsidence of UFT/SIN air masses (Fig. 6). PC2 primarily separated air masses with low $q$, high $O_3$, and rather high $CH_4$ mixing ratios from air masses with opposite characteristics in the winter half year. On its own, PC2 was a less reliable indicator for vertical transport because low $q$ and high $O_3$ mixing ratios

suggested a subsidence of UFT/SIN air masses while high $CH_4$ mixing ratios suggested an uplift of ML air masses. Possibly, PC2 reflects not only vertical but also horizontal concentration gradients.

In the summer half year, the loading of $CO_2$ on PC1 had the opposite sign compared to the loadings of $CH_4$ and CO on PC1. This observation suggests that low $CO_2$ mixing ratios were generally indicative for ML air masses during the vegetation period due to the $CO_2$ removal by photosynthesis. Additionally, $p_{GAP}$ contributed more strongly to the first two PCs than

in the winter half year and was anti-correlated with $q$. $\Delta\theta_v$ exhibited very small loadings on the first two PCs in the periods April–May and June–July, indicating a poor correlation with the other variables.

In April and May, an uplift of ML air masses was indicated by low scores of PC1, which primarily reflected high $q$, low $CO_2$ mixing ratios, and low $p_{GAP}$, and by low scores of PC2, which primarily reflected high CO and $CH_4$ mixing ratios.

In June and July, PC1 separated cases with high $CH_4$, low $CO_2$, and low $p_{GAP}$, which indicated an uplift of ML air masses,

from cases with opposite characteristics, which indicated a subsidence of UFT/SIN air masses. In the same period, PC2 could not be interpreted unambiguously with respect to vertical transport because high CO mixing ratios, which suggested ML air masses, but also low $q$ and high $p_{GAP}$, which suggested UFT/SIN air masses, contributed to low scores of PC2.

In August and September, PC1 separated cases with high CO and $CH_4$, low $CO_2$, and high $\Delta\theta_v$ (i.e. low static stability), which was typical for ML air masses, from cases with opposite characteristics, which was typical for UFT/SIN air masses.

Again, PC2 was a less reliable air mass indicator because low $p_{GAP}$ and high $q$, which suggested an uplift of ML air masses, but also high $CO_2$ mixing ratios, which were typical for UFT/SIN air masses, increased the scores of PC2 in August and September (Fig. 6).

## 3.2   Air mass regimes in February and March

The interpretation of the air mass regimes, which were confined by the 33.3 % and 66.6 % percentiles of the first two PCs,

differed among the 2-month periods except for the periods December–January and February–March. As an example, the period February–March is discussed in detail using summary statistics. A case study illustrating the classification results in the measured time series is shown in the supplement Sect. S2. The air mass regimes were named as "x,y", where x and y are prominent features of PC1 and PC2, respectively. In the period February–March, x was called "polluted", "moderately polluted" (mod. poll.), or "unpolluted" with respect to the pollutants CO, $CH_4$, and $CO_2$ while y was called "high $q$", "intermediate $q$"

(interm. $q$), or "low/interm. $q$". The joint data availability of the PCA input variables was 87 % of the time in February–March.

For the regimes "polluted, high $q$", "polluted, interm. $q$", and "polluted, low/interm. $q$", the summary statistics of all seven PCA input variables were consistent with a recent uplift of ML air masses (Fig. 7a–g). Compared to the other regimes, CO,




$CH_4$, and $CO_2$ generally exhibited high mixing ratios and $q$ was also relatively high while $O_3$ and $p_{GAP}$ were low or interme-diate. $\Delta\theta_v$ indicated a weakly stable stratification with a median of approximately $-8$ K for the "polluted" regimes.

The other chemical and meteorological measurements supported the interpretation of the "polluted" regimes (Fig. 7h–r). Relatively high values for rH (medians between 80 % and 90 %) and total precipitation were consistent with a current uplift of air masses. For $NO_y$ and $NO_x$, local pollution events ($\sigma_{NO_x} > 0.4$ ppb) were excluded while for [7]Be, $N_{90}$, eBC, and $PM_{10}$,

cases with precipitation at the ZPLT were excluded from the summary statistics because local emissions or wet deposition obscure the chemical signature resulting from vertical transport processes. Nevertheless, the data availability of these variables was 75 % to 78 % of the time. The "polluted" regimes exhibited high $NO_y$, $NO_x$, and [222]Rn and low [7]Be concentrations compared to the other regimes, which underlines the strong influence of the ML. The high $NO_x$ mixing ratios suggest a strong influence of combustion processes. High [222]Rn concentrations are typical for the continental ML because of natural emissions

from the ice-free land surface (Griffiths et al., 2014). [7]Be concentrations are generally low in the ML due to the formation by cosmic rays in the stratosphere and upper troposphere (Stohl et al., 2000). The aerosol concentrations $N_{90}$, eBC, and $PM_{10}$ were somewhat elevated for the "polluted" regimes but not as strongly as the gas-phase measurements. This finding could be due to wet deposition during the uplift of the air masses. At ZSG, southeasterly to southerly winds with varying velocities were frequently observed for the three regimes under consideration (see supplement, Fig. S1a,d,g). The observations suggest that an

uplift of ML air masses was mostly caused by a low pressure system or a south foehn event in February and March.

The regimes "mod. poll., high $q$", "mod. poll., interm. $q$", and "mod. poll., low/interm. $q$" exhibited intermediate CO, $CH_4$, and $CO_2$ mixing ratios and an intermediate $p_{GAP}$ (Fig. 7a–g). For these air masses, vertical transport processes were not inferable but the intermediate mixing ratios suggest an influence from both the ML and the free troposphere. Thus, the three regimes under consideration were attributed to the air mass class HYBRID, which was in line with intermediate $NO_y$, $NO_x$,

and [222]Rn concentrations (Fig. 7l–n).

The regime "unpolluted, high $q$" was generally characterized by the lowest CO, $CH_4$, and $CO_2$ mixing ratios and a strongly stable stratification (median $\Delta\theta_v$ of $-15$ K). However, low $O_3$ mixing ratios, high $q$, and an intermediate and strongly variable $p_{GAP}$ indicated that the air masses did not originate from the upper troposphere or stratosphere but from the lower troposphere or the marine boundary layer (Fig. 7a–g). This interpretation was in line with intermediate [7]Be and [222]Rn concentrations

and low $NO_y$, $NO_x$, $N_{90}$, eBC, and $PM_{10}$ concentrations (Fig. 7l–r). Intermediate rH with a median of 54 % corroborated some recent influence of the free troposphere. The air temperature was high compared to the other regimes (Fig. 7i), the wind direction at ZSG almost always had a south component, and the wind velocity at ZSG showed the highest mean of $10.22\,\mathrm{m\,s^{-1}}$ among the regimes (see supplement, Fig. S1c). The summary statistics did not allow for drawing a final conclusion on the influence of the marine boundary layer and thus, the regime "unpolluted, high $q$" was attributed to the air mass class HYBRID

and the subclass MBL/UFT unless the criteria for LRMD were fulfilled.

Balzani Lööv et al. (2008) clustered back-trajectories for the Jungfraujoch, another mountain site in the Alps, and demon-strated that some air masses originated from the marine boundary layer above the tropical Atlantic Ocean and were transported to the Jungfraujoch inside the free troposphere within 5 to 15 days, resulting in rather high rH (average of 67 %), high temper-



atures, and low $O_3$, CO, $NO_y$, and $NO_x$ mixing ratios. Air masses with similar trajectories may also reach Mt. Zugspitze and represent, at least partly, the regime "unpolluted, high $q$".

For the regimes "unpolluted, interm. $q$" and "unpolluted, low/interm. $q$", the CO, $CH_4$, and $CO_2$ mixing ratios were generally low but not as low as for the regime "unpolluted, high $q$". $O_3$ was intermediate for the regime "unpolluted, interm. $q$" and high for the regime "unpolluted, low/interm. $q$". Together with a strongly negative $\Delta\theta_v$ (median of approximately $-15$ K), i.e. a

strongly stable stratification, and a generally high $p_{GAP}$, these observations indicated a subsidence of UFT/SIN air masses for the two regimes under consideration unless the criteria for LRMD were fulfilled (Fig. 7a–g). This interpretation was supported by a generally low rH (medians of 45 % and 25 %), a total precipitation of almost zero, low $NO_y$, $NO_x$, $N_{90}$, eBC, and $PM_{10}$ concentrations, and high $^7$Be concentrations compared to the other regimes (Fig. 7h–r).

For all regimes except "polluted, low/interm. $q$", the boxplots of global radiation ($R_g$) were similar with medians close to

zero (Fig. 7k), indicating that the air mass characteristics were generally independent from day- and nighttime in February and March.

### 3.3 Seasonal frequencies of the air mass classes

In the entire year, the data availability limited the percentage of classifiable cases to 78% of the time. Data gaps occurred predominantly in the periods June–July and October–November (Fig. 8). On average, ML, UFT/SIN, and HYBRID air masses

accounted for 31 %, 14 %, and 55 % of the classifiable cases, respectively. The percentage of UFT/SIN air masses is in reasonable agreement with the study of Yuan et al. (2019), in which 13.6 % of the long-term $CO_2$ data from the UFS were selected as baseline concentrations using an univariate statistical approach called "adaptive diurnal minimum variation selection". For earlier $CO_2$ data from ZSG and a pedestrian tunnel, approximately 70 m above the UFS, the percentage of baseline concentrations was 19.5 % and 9.9 %, respectively, according to the same study.

The air mass class ML was attributable to three of the nine air mass regimes in all 2-month periods except June–July and October–November when the class ML comprised two regimes (Fig. 8). This finding implies that air masses are lifted up to the UFS with similar frequencies in all seasons although the underlying processes may vary. However, the frequency of the air mass classes was constrained to some extent by using the 33.3 % and 66.6 % percentiles of the first two PCs as thresholds between the air mass regimes. These percentiles were somewhat arbitrary but allowed for a distinction between more unambiguous air

masses at both edges of the distribution and more ambiguous air masses in the middle part of the distribution.

UFT/SIN air masses were attributable to two regimes in the winter half year but only to one regime in the period August–September and to none of the regimes in the periods April–May and June–July. UFT/SIN air masses were only evident if the summary statistics of an air mass regime indicated a dominating influence of subsidence. From April to July, subsidence appeared to be very rare at the UFS so that UFT/SIN air masses could not be isolated by splitting the data at the 33.3 % and

66.6 % percentiles of the first two PCs. This result can be explained by the fact that subsidence in high-pressure systems is counteracted by thermally induced uplift in the daytime, especially in the summer half year. However, the data availability in June–July was only 56 %, which questions the long-term representativeness for this period. Other statistical classifications at the high-alpine sites Jungfraujoch and Mt. Bachelor Observatory indicated that air masses of the undisturbed free troposphere





occur year-round but less frequently in summer than in winter (Herrmann et al., 2015; Ambrose et al., 2011). In principle, these air masses do not have to descend but can also be advected horizontally to high-alpine sites. Such cases may be included in the class HYBRID and most likely in the subclass MBL/UFT.

HYBRID air masses including MBL/UFT comprised four to five regimes from August to March and even six and seven regimes in the periods April–May and June–July, respectively. In most of the 2-month periods, MBL/UFT air masses accounted for one regime (Fig. 8). Seven LRMD events were identified in the whole year, predominantly from April to July, which is in line with the study of Flentje et al. (2015) that found 5 to 15 Sahara dust events per year at the mountain site Hohenpeissenberg, approximately 40 km north of Mt. Zugspitze.

### 3.4 Comparing the statistical and mechanistic classifications

The ceilometer-based MLH at GAP exhibited a pronounced diurnal cycle with a maximum in the late afternoon or early evening on days with a high total global radiation (not shown). Little diurnal variations of the MLH were observed for days with a low total global radiation. Thermally induced anabatic winds were most frequently observed during summer and daytime whereas thermally induced katabatic winds occurred most frequently at night and in the first few hours after sunrise (not shown). The patterns described above were expected and confirmed the plausibility of the mechanistic approach.

When using the mechanistic approach for validation purposes, the $MLH_{GAP}$ was discarded if clouds were detected at a height of $< 4$ km a.s.l. because low level clouds increase the uncertainty of the MLH retrieval. Finally, 25 % of the statistically classifiable cases were evaluated with the mechanistic approach. In 3 % of the statistically classifiable cases, the mechanistic approach suggested an influence of the local ML on the UFS, mostly due to thermally induced anabatic winds. It should be noted that the mechanistic approach is not able to detect synoptic uplift processes because low-level clouds and precipitation result in data gaps of the $MLH_{GAP}$. In 5 % of the statistically classifiable cases, katabatic winds and a low $MLH_{GAP}$ compared to the UFS level suggested non-ML air masses. In another 17 % of the statistically classifiable cases, the winds were not thermally induced but the $MLH_{GAP}$ suggested non-ML air masses.

When the mechanistic approach suggested ML air masses, the statistical approach agreed in 54 % of the cases and yielded HYBRID air masses in almost all other cases (Fig. 9). This finding indicates a poor performance of the statistical approach in the case of thermally induced uplift. On the other hand, thermally induced uplift only accounts for a small fraction of the time according to the mechanistic approach. Other uplift processes including non-local uplift were not identifiable with the applied mechanistic approach. LRMD was found in 3 % of the cases, which were mechanistically classified as influenced by the ML, likely due to an overestimation of the ceilometer-based $MLH_{GAP}$ if the advected dust layer merged with the ML.

In the presence of katabatic winds and a low $MLH_{GAP}$ compared to the UFS level, the statistical classes UFT/SIN and HYBRID were both considered to agree with the mechanistic approach (illustrated by the thick black line in Fig. 9). Thus, the level of agreement was 91 % for katabatic winds and relatively low $MLH_{GAP}$, whereby UFT/SIN and HYBRID air masses accounted for half of these cases, respectively. When the $MLH_{GAP}$ was lower than the UFS level but the winds were not thermally induced, the level of agreement between the classifications was a bit lower (83 %) with UFT/SIN and HYBRID air masses accounting for 31 % and 52 %, respectively.





Uncertainties of the mechanistic approach arise mainly from the assumption of identical MLHs at the UFS and GAP and the lack of information on non-local uplift followed by horizontal advection of air masses. Uncertainties of the statistical approach result primarily from the lack of objective thresholds between the air mass classes and a varying significance of sources and sinks of the gases used. In view of these uncertainties, a reasonable agreement between the approaches was found when the mechanistic approach indicated non-ML air masses.

## 3.5   Implications for real-time operational mode

To classify air masses in real-time, the scores of the first two PCs can be approximated by using the loadings that were calculated in this study. Thus, the same thresholds as described here can be applied to determine the regime and the class of the current air mass. To standardize the original variables, the arithmetic mean and standard deviation of the current 2-month period are required but these statistics are only known at the end of the 2-month period. Due to the long-term trends of $CO_2$, $CH_4$, and
CO and the interannual variability of all variables, it is advisable not to use the 2-month means and standard deviations from the year investigated in the present study but to estimate these statistics from recent multi-year records. To make use of the known part of the current 2-month period, the mean ($\mu$) and standard deviation ($\sigma$) could be updated regularly using weighted averages

$$\mu = (1 - f)\mu_l + f\mu_c \qquad (6)$$

$$\sigma = (1 - f)\sigma_l + f\sigma_c \qquad (7)$$

where $\mu_l$ and $\sigma_l$ are the 2-month mean and standard deviation, respectively, estimated from the long-term record, $\mu_c$ and $\sigma_c$ are the mean and standard deviation of the known part of the current 2-month period, and $f$ is the time fraction that is known in the current 2-month period. In real-time operational mode, additional uncertainties can arise from uncorrected measurement
artifacts.

## 4   Conclusions

In this study, a novel statistical approach was developed to discriminate between the air mass classes ML (recent contact with the mixing layer), UFT/SIN (undisturbed free troposphere or stratospheric intrusion), and HYBRID (influences of both the mixing layer and the free troposphere or ambiguous characteristics) at the Schneefernerhaus at Mt. Zugspitze. A main
purpose of the classification scheme is a later use in real-time operational mode. The scheme was based on the first two principal components, which were calculated from five gas-phase (CO, $CH_4$, $CO_2$, $O_3$, $q$) and two meteorological ($\Delta\theta_v$, $p_{GAP}$) variables but differed between the six 2-month periods. Additionally, the $PM_{10}$ concentration and its 30 min standard deviation were needed to identify long-range transport of mineral dust. In retrospect, local pollution events on the mountain were evident from a high 30 min standard deviation of $NO_x$ in approximately 9 % of the time. But these events are too short
to be predictable on a 30 min basis in real-time applications and are thus neglected by the classification scheme.





While the first principal component was a suitable air mass indicator throughout the year, the second principal component left room for ambiguities. ML air masses were detected in all seasons with similar frequencies (average of 31 % of the classifiable cases). UFT/SIN air masses were predominantly found in the winter half year (23 % of the classifiable cases from October to March) but subsidence was so rare from April to July that the class UFT/SIN was not determinable in these months. HYBRID air masses (average of 55 % of the classifiable cases) mostly exhibited intermediate characteristics leaving room

for ambiguities. For 17 % of HYBRID, it remained unclear whether the air mass originated from the lower undisturbed free troposphere or the marine boundary layer. To achieve a distinction between these two areas of origin, future work with trajectory models is necessary. 5 % of HYBRID was explained by long-range transport of mineral dust.

Independent chemical and standard meteorological measurements such as $NO_y$, $^{222}Rn$, and precipitation were in line with the statistical approach. A mechanistic classification based on ceilometer and standard meteorological measurements was

feasible in 25 % of the statistically classifiable cases and predominantly suggested non-ML air masses in these cases, which was in good agreement with the statistical approach. In the rare cases of thermally induced uplift, however, the statistical approach often misclassified ML air masses as HYBRID.

In principle, the statistical classification scheme can be used in real-time operational mode if the unknown arithmetic means and standard deviations of the observational variables in the current 2-month period can be estimated with sufficient accuracy.

Future work should test the real-time applicability and quantify the uncertainty arising from the use of approximated means and standard deviations for standardization.

The framework of the presented statistical classification might also be useful at other high-alpine sites because it is based on common measurements.

*Data availability.* The $^{222}Rn$ data are available from the World Data Centre for Greenhouse Gases at https://gaw.kishou.go.jp/search/file/

0019-6031-6002-01-01-9999 after free registration. The other data can be requested from the authors.

*Author contributions.* AS, KF, CT, and TR developed the research concept and design. LR, CS, and AM collected measurement data and controlled the data quality. AS performed the data analysis. AS prepared the manuscript with contributions from all co-authors.

*Competing interests.* The authors declare that they have no conflict of interest.

*Acknowledgements.* This study was supported by the Bavarian State Ministry for the Environment and Consumer Protection in the framework

of the project PureAlps (VH-ID: 70667 / TNT01 NaT-70667) and was performed in close coordination with the Bavarian Environment Agency. In this work, data from climate stations, which were set up during the project KLIMAGRAD (funded by the Bavarian State Ministry for the Environment and Consumer Protection, ZKL01Abt7_18458) were used. We thank Germany's National Meteorological Service for



providing meteorological data from the sites ZSG, UFS, and GAP, ceilometer measurements, $Rn^{222}$, and $^7Be$ data. Meteorological data from the UFS were provided through the Alpine Environmental Data Analysis Centre (AlpEnDAC.eu). We thank the AlpEnDAC staff for their support. Henry Schmölz (Bavarian Avalanche Warning Service) is acknowledged for providing meteorological data for the site ZPLT. We thank Markus Hermann and his team (Leibniz Institute for Tropospheric Research) for providing the SMPS data. Ralf Sohmer and Cedric Couret (German Environment Agency) are acknowledged for their assistance in the measurements and quality control of the trace gases.
5    We thank Karl Lapo for his helpful comments on the manuscript. Manfred Kirchner and Matthias Mauder are acknowledged for valuable discussions.



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





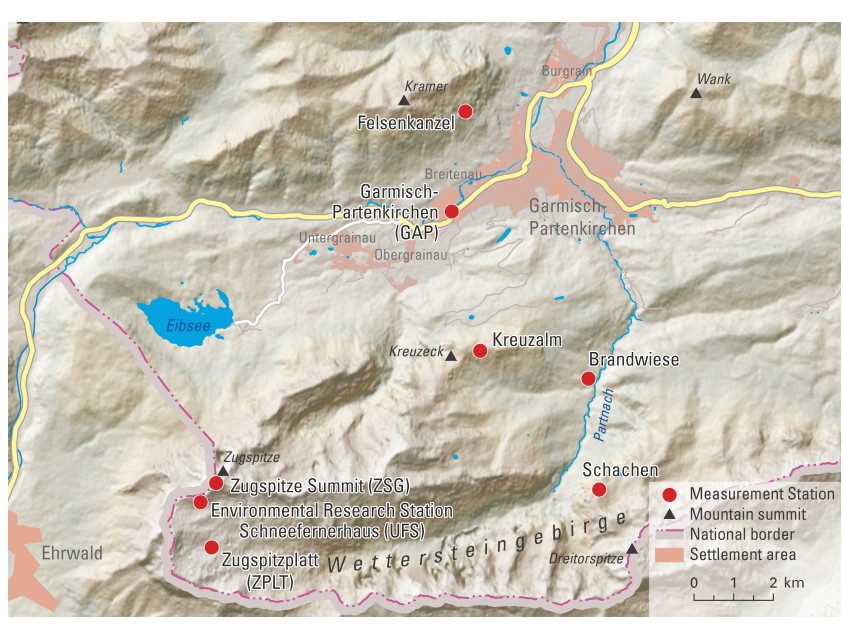

**Figure 1.** Location of the measurement sites in the surrounding of Mt. Zugspitze. North is at the top.





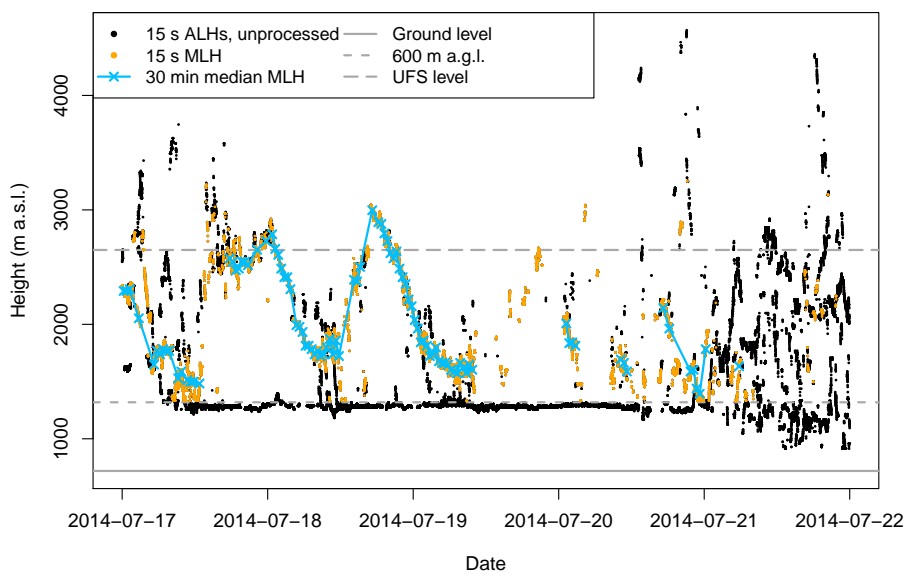

**Figure 2.** Post-processing of the ceilometer-based aerosol layer heights (ALHs), which were used to determine the mixing layer height (MLH) at GAP, illustrated for the period from 17 to 21 July 2014. From the MLH, outliers have been removed. Points of the 30 min median MLH were connected by a line if separated by maximum 3 h.

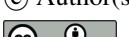



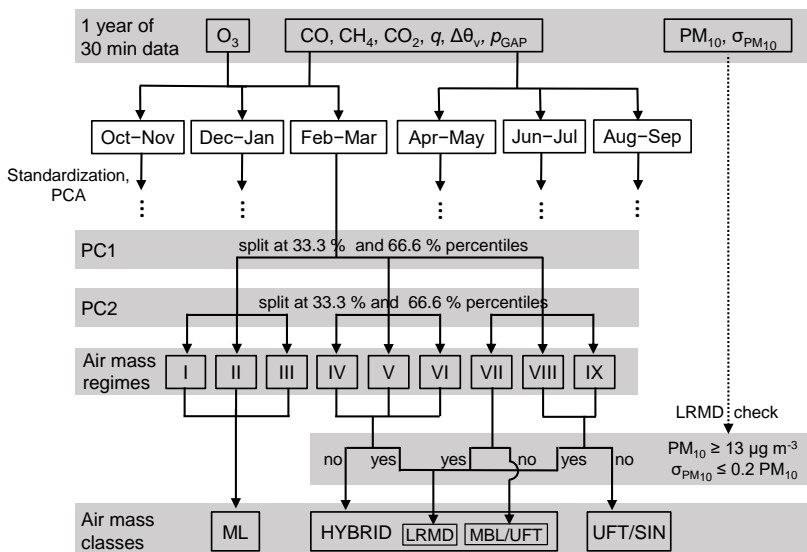

**Figure 3.** Scheme for statistical air mass discrimination at the UFS using the first (PC1) and second principal components (PC2) of a 6- or 7-dimensional data set in each 2-month period, as illustrated for February and March. ML denotes a recent contact with the mixing layer, UFT/SIN includes the undisturbed free troposphere and stratospheric intrusions, and HYBRID reflects influences from both ML and UFT/SIN and includes the subclasses LRMD (long-range transport of mineral dust) and MBL/UFT (influence of the marine boundary layer or UFT).



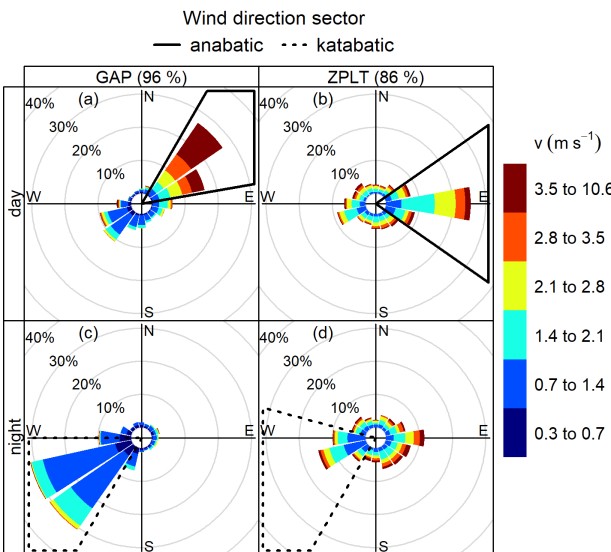

**Figure 4.** Wind rose plots showing the frequency of counts by wind direction and velocity (v) at the sites GAP (a,c) and ZPLT (b,d) during day- (a,b) and nighttime (c,d) in summer (June to August). Daytime was defined by a global radiation of $> 5\,\mathrm{W\,m^{-2}}$ at minimum one of the sites used in this study; all other cases were treated as nighttime. The plots were used to define wind direction sectors for thermally induced anabatic and katabatic winds. The percentage in brackets specifies the data availability.




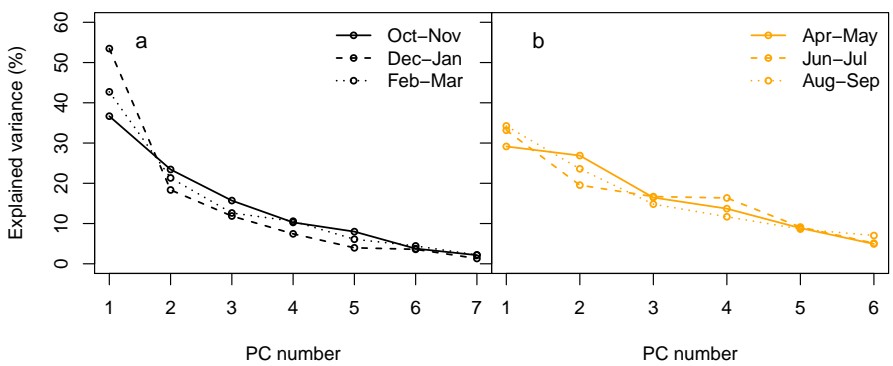

**Figure 5.** Percentage of explained variance as a function of the principal component (PC) number for each 2-month period: (a) winter half year and (b) summer half year.





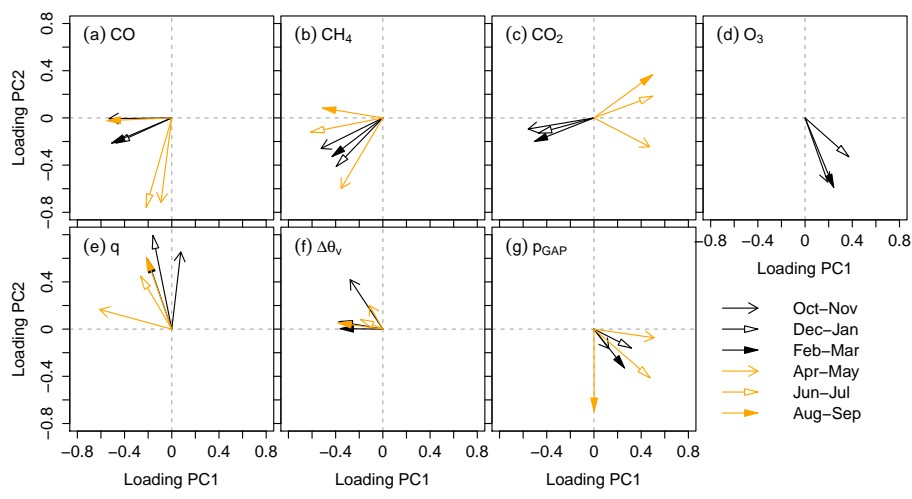

**Figure 6.** Loadings of the input variables on the first (PC1) and second principal component (PC2) for each 2-month period. $O_3$ was only used as an input variable in the winter half year (black arrows).





**Figure 7.** Summary statistics of PCA input variables (a–g) and validation variables (h–r) for the nine statistical air mass regimes in February and March 2014. The boxes show the quartiles and the boxwidth is proportional to the number of data points; each whisker is limited to a length of 1.5 times the interquartile range. Total precipitation ($\text{precip}_{\text{ZPLT}}$) is depicted in a bar plot. Unless labeled differently, the data was measured at the UFS except for $\Delta\theta_v$ that represents the maximum difference among the sites. Symbols are explained in Sect. 2.2 and 2.3. Above the plots, the data availability is specified as fraction of time. For $NO_x$ and $NO_y$, local pollution events were excluded. For $^7\text{Be}_{\text{ZSG}}$, $N_{90}$, eBC, and $PM_{10}$, cases with precipitation ($\text{precip}_{\text{ZPLT}} > 0$) were excluded. The axis of ordinate was limited to $\mu \pm 3\sigma$, where $\mu$ and $\sigma$ are mean and standard deviation of individual regimes, respectively.





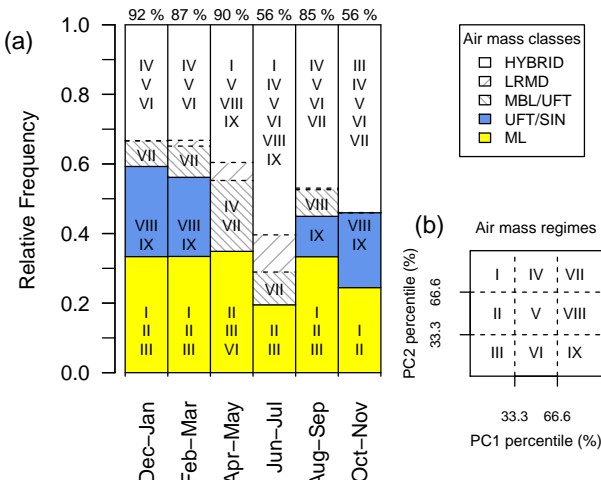

**Figure 8.** (a) Relative frequency of the air mass classes as fraction of the classifiable cases in each 2-month period. Above the plot, the percentages of classifiable cases are given as fraction of time. (b) Numbering of the nine underlying air mass regimes. LRMD (long-range transport of mineral dust) represents certain cases within the regimes that were not attributed to ML (recent contact with the mixing layer). LRMD and MBL/UFT (influence of the marine boundary layer or UFT) are part of HYBRID (influences of both the mixing layer and the free troposphere or ambiguous). UFT/SIN denotes undisturbed free troposphere or stratospheric intrusion.




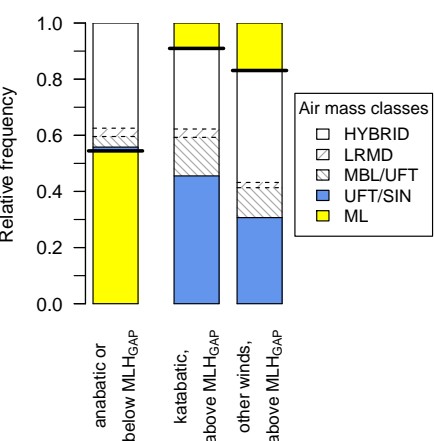

**Figure 9.** Comparison between the statistical and mechanistic classifications of air masses. The mechanistic classes were based on the mixing layer height at GAP ($MLH_{GAP}$) and thermally induced anabatic and katabatic winds and accounted for 3 %, 5 %, and 17 % of the statistically classifiable cases (from left to right). The thick black lines show the percentages considered as an agreement between the approaches. $MLH_{GAP}$ was discarded if clouds were detected below 4 km a.s.l..



**Table 1.** Chemical measurements used in this study. The given interval is the temporal interval at which the data were provided by the respective institution. UV denotes ultraviolet light. The symbols for the measured quantities are explained in the text.

| Symbol | Unit | Interval | Instrument | Measurement principle | Institution |
|---|---|---|---|---|---|
| CO | ppb | 1 min | AL5002, Aero-Laser, Garmisch-Partenkirchen, Germany | UV fluorescence | UBA[e] |
| $CH_4$ | ppb | 1 min[a] | EnviroSense 3000i, PICARRO, Sunnyvale, USA | cavity ringdown spectroscopy | UBA[e] |
| $CO_2$ | ppm | 1 min[a] | EnviroSense 3000i, PICARRO, Sunnyvale, USA | cavity ringdown spectroscopy | UBA[e] |
| $O_3$ | ppb | 1 min | Model 49i, Thermo Scientific, Waltham, USA | UV absorption | UBA[e] |
| $NO_y, NO_x$ | ppb | 1 min | CraNOx II, ECO PHYSICS, Dürnten, Switzerland | chemiluminescence | UBA[e] |
| HCHO | ppb | 10 min | AL4021, Aero-Laser, Garmisch-Partenkirchen, Germany | fluorometric Hantzsch reaction[c] | TUM[f] |
| $^{222}Rn$ | $Bq\ m^{-3}$ | 2 h | Radon sphere 270, IGU, Wörthsee, Germany | electrostatic deposition followed by alpha particle spectrometry[d] | DWD[g] |
| $^7Be$ | $mBq\ m^{-3}$ | 12 h | Glass fiber filter and pump, Tracerlab, Köln, Germany | gamma spectrometry | DWD[g] |
| $dN\,(dlog\,d_p)^{-1}$ | $cm^{-3}$ | 10 min | SMPS[b] model 3936, TSI Inc., Shoreview, USA | separation of charged particles in an electric field | TROPOS[h] |
| $PM_{10}$ | $\mu g\ m^{-3}$ | 1 min | FH 62 C14, Thermo Scientific, Waltham, USA | attenuation of beta radiation | UBA[e] |
| eBC | $\mu g\ m^{-3}$ | 1 min | MAAP[b] model 5012, Thermo Scientific, Waltham, USA | light attenuation and reflection by particle-laden quartz fiber filters | UBA[e] |

[a] 30 min in the year 2014; [b] see Birmili et al. (2016) and Sun et al. (2018) for details on the Scanning Mobility Particle Sizer (SMPS) and the Multi Angle Absorption Photometer (MAAP); [c] see Leuchner et al. (2016) for details; [d] see Steinkopff et al. (2012) for details; [e] German Environment Agency; [f] Ecoclimatology, Technical University of Munich; [g] German Meteorological Service; [h] Experimental Aerosol and Cloud Microphysics, Leibniz Institute for Tropospheric Research





**Table 2.** Criteria for thermally induced anabatic and katabatic winds occurring simultaneously at GAP and ZPLT. $\varphi$ is wind direction, $v$ is wind velocity, and $\Delta\theta_v$ is the range of the pseudo-vertical profile of virtual potential temperature.

| Wind class | $\varphi_{\mathrm{GAP}}$ | $\varphi_{\mathrm{ZPLT}}$ | $\Delta\theta_v$ | $v_{\mathrm{ZPLT}}$ and $v_{\mathrm{GAP}}$ | Duration |
|---|---|---|---|---|---|
| Anabatic | 30° to 80° | 55° to 125° | $> -8\,\mathrm{K}$ | | $\geq 1\,\mathrm{h}$ |
| Katabatic | 210° to 270° | 210° to 285° | $\leq -8\,\mathrm{K}$ | $< 3\,\mathrm{m\,s^{-1}}$ | $\geq 1\,\mathrm{h}$ |