# Peer review of "Multivariate statistical air mass discrimination for the high-alpine observatory at the Zugspitze mountain, Germany"

_Atmospheric Chemistry and Physics, 2019_

## Referee Comment (RC1) · Anonymous Referee #2 · 26 Apr 2019

Manuscript ACP-2019-211
**Multivariate statistical air mass discrimination for the high-alpine observatory at the Zugspitze mountain, Germany**

Introduction

The objective of this study is to devise a method for real-time air mass classification based on routine chemical and meteorological measurements at high-altitude sites. Compared to existing classification approaches, which typically consider the concentration of single chemical constituents or at most the ratio of two concentrations, this method adopts a multivariate approach based on principal component analysis (PCA). The study considers one year of nearly continuous measurements. Based on PCA scores, nine different "air mass regimes" are identified. In turn, these regimes are mapped to three "air mass classes", respectively mixed-layer air, free tropospheric/stratospheric air, and air with hybrid characteristics. The mapping from air mass regimes to air mass classes is not rigidly prescribed, it varies seasonally, and it introduces a subjective element into the classification. The results of the statistical classification scheme are compared to those of a so-called "mechanistic approach", where individual cases are classified based on meteorological measurements and a-priori knowledge of local wind circulation patterns. The comparison highlights some limitations of the proposed statistical classification scheme.

Recommendation

The manuscript is well organized, well written and appropriately concise. Figures are of good quality. The subject matter is well within the thematic scope of ACP. I have the feeling that some characteristics and implications of the proposed classification method have not been fully considered (see in particular comments 1 and 2 below). However, given the originality and novelty of the core idea (classifying air masses using a multi-variate method based on chemical and meteorological measurements) the manuscript is probably a good candidate for publication. A request for revisions is recommended.

General comments

(1)  Pre-processing of the data before use in the statistical classification method is limited to standardization (that is, adjustment of the sample mean to 0 and of the sample variance to 1). I am wondering whether any slightly more sophisticated pre-processing could be beneficial.
For instance:
(a) Some of the variables in the data matrix have well-defined seasonal and diurnal cycles. Would it be possible to determine average annual and daily cycles, and to remove them from the data set? Performing the analysis on deviations from the average cycles might improve classification results.
(b) PCA does not require the data to follow multivariate normal distributions, but its results can often be interpreted more easily if they do. It strikes me that most variables are concentrations, therefore their PDF will certainly be markedly non-Gaussian. Would a cleverly designed variable transformation allow bringing more variance into the leading principal components?

(2)  The matching between air-mass regimes (I-IX) and air-mass classes (ML, UFT/SIN, HYBRID) is different in each two-month period (see Figure 8). The manuscript text contains little or no information about the overarching logic. Why was this necessary? What criteria were used to attribute regimes to classes, how did these criteria change with the season?
In my opinion, the ad-hoc tuning of the method is a serious shortcoming. It is clearly a subjective component of the classification, and as such it cannot be exported to other sites. The authors do not explain this point in a satisfactory manner, and they probably should. Why wasn't it possible to design a fully objective classification rule? Formal methods to identify classification rules exist and could be used (see for instance chapter 14 in Wilks, 2011, *Statistical Methods in the Atmospheric Sciences*. DOI: 10.1016/B978-0-12-385022-5.00014-2).

Specific comments

(3)     Note: the line numbering in pages 2-end is wrong, i.e., the 6[th] line from the top is labelled as "5" and so on, as if the the first line were 0. In what follows, I'm using this unusual "zero-based" system.

(4)     Nomenclature. The first air-mass class is labelled ML, for "mixing layer". I'm wondering if this is appropriate. A mixed layer, by definition, has nearly adiabatic lapse rate. The boundary layer (BL) is not always well-mixed, especially at night. On page 1, line 12, it is stated that "the terms ML and BL can be defined synonymously...". In my mind, the two concepts are quite distinct. I'd rather say that the BL might sometimes include a ML. I don't really have a strong opinion on this matter, but anyway I suggest renaming the first air-mass class to BL, for "boundary layer".
A similar comment applies to "mixing layer height" (MLH). This should probably become "boundary layer height" (BLH), in particular because, according to the description of the wavelet detection algorithm, MLH/BLH potentially includes multiple aerosol layers. These typically develop in connection with inversion layers, i.e., non-mixed parts of the atmosphere.

(5)     Page 1, title. "Discrimination" or "classification"? The two terms have slightly different meanings. See again chapter 14 in Wilks 2011.

(6)     Page 1, lines 8 and 12. Use of the word "classifiable". I believe these statements should be formulated more clearly. As they are now, they seem to allude to the intrinsic "ability" of the methods to separate the events, and seem to suggest that the statistical method permits to obtain a meaningful classification much more often than the mechanistic one (78% of the time as opposed to 25%). Instead, these two percentage only represent the availability of the input data for the two methods. Please use something like: "Due to data gaps, only x% of the investigated year could be classified".

(7)     Page 2, lines 10-11. Foehn flows are listed among processes that cause air mass *lifting*. This is inexact and quite confusing. Foehn is a fall wind: its dynamics are inextricably tied to air-mass *descent* (not lifting!) on the lee side of a mountain range. That said, intense foehn certainly causes mechanical mixing of the lower atmosphere, which may result in transport of chemical species from the PBL to the free troposphere. Zellweger et al (2003, cited in the manuscript) list foehn among the meteorological conditions in which free-troposphere air masses *are mixed* with PBL air masses (page 781, top of second column). Correctly, Zellweger et al (2003) do not mention "lifting" in relation to foehn. Please revise. The comment also applies to page 2, line 23 and to page 12, line 15.

(8)     Page 2, line 30. "… because the MLH is a meteorological quantity". Wording could be more careful here. MLH/BLH is not a directly measured quantity, but rather an estimate obtained from measurements of other quantities. The determination of MLH/BLH from a vertical profile can be quite tricky, too. I'd rather say something like: "… because determination of the MLH from vertical profiles of measured quantities requires a-priori meteorological knowledge".

(9)     Page 5, line 17. "...set zero" → "...set to zero".

(10)    Page 5, line 21. Please delete the blank space before the full stop.

(11)    Page 6, line 17. "...using the Clausius Clapeyron equation". Or rather a numerical approximation? There are many such formulas: Goff-Gratch, Magnus-Tetens, Bolton… which one?

(12)    Page 6, line 19. I think the standard notation should be either $e^x$ or $\exp\{x\}$. Also, please replace $T_v$ by $\overline{T}_v$, to indicate the vertical averaging.

(13)    Page 7, line 2. I personally find the sign convention counterintuitive. Although static stability conventionally corresponds to $d\theta/dz > 0$, here $\Delta\theta$ is greater than zero when $\theta$ decreases with height. Why not computing $\Delta\theta/\Delta z$?

(14)    Page 7, lines 20-21. "The MLH attribution was based on the idea that the MLH varies only gradually". I am not sure this is always appropriate over mountains. Horizontal advection of aerosol layers due to mountain venting can cause spatial and temporal discontinuities in MLH.

(15)    Page 10, lines 9-10. Why using –8 K to discriminate "weak" and "strong" static stability?

(16)    Page 11, line 29. Are $CH_4$ and $CO_2$ pollutants?

---

## Short Comment (SC1) · 19 May 2019

**My responses are marked in *italic and blue* and were directly inserted below each comment of the referee.**

*I thank Referee #1 for the valuable comments, which help to improve the manuscript. In the following, I reply to the general comments, which offer the largest potential for discussion.*

[Figure]

**General comments from Referee #1**

(1) Pre-processing of the data before use in the statistical classification method is limited to standardization (that is, adjustment of the sample mean to 0 and of the sample variance to 1). I am wondering whether any slightly more sophisticated pre-processing could be beneficial.
For instance:
(a) Some of the variables in the data matrix have well-defined seasonal and diurnal cycles. Would it be possible to determine average annual and daily cycles, and to remove them from the data set? Performing the analysis on deviations from the average cycles might improve classification results.

*In our analysis, the seasonal course was roughly removed by subtracting the 2-month mean in each of the 2-month periods when standardizing the data. In principle, it would be beneficial to remove the seasonal course more accurately so that the leading principal components would only depend on the shorter-term variability of the time series. In the beginning of the data analysis, we considered the determination and removal of the seasonal course with a spectral approach such as a wavelet filter. However, this idea was abandoned because it requires knowledge of the time series within a window centered at the time of interest and is thus not applicable to real-time operational mode. It would be possible to remove an average seasonal cycle from each variable based on a larger, multi-year data set. Due to interannual variability, however, this approach would only partly remove the seasonal course. For example, Scheel et al. (1999) showed that the monthly mean $O_3$ mixing ratios at Mt. Zugspitze can strongly differ between individual years and the 10-year ensemble. Therefore and because a large fraction of the seasonal variability is already removed by subtracting the 2-month mean values, I doubt that the removal of an average seasonal cycle would substantially improve the classification results.*

*The mean diurnal cycles of the analyzed variables reflect shifts between air masses*

*that we aim to identify. Afternoon maxima of atmospheric constituents such as water vapor, CO, $^{222}Rn$, and $NO_y$ at high-alpine sites have been explained by thermally in-duced uplift processes including anabatic winds (Forrer et al. (2000); Zellweger et al. (2003); Griffiths et al. (2014)). These conditions are typically accompanied by after-noon minima of atmospheric stability and air pressure. The latter is associated with plain to mountain winds in the Northern Alps (Lugauer and Winkler, 2005). Removing the average diurnal cycle would complicate the classification of air masses in the case of thermally induced vertical transport.*

(b) PCA does not require the data to follow multivariate normal distributions, but its results can often be interpreted more easily if they do. It strikes me that most vari-ables are concentrations, therefore their PDF will certainly be markedly non-Gaussian. Would a cleverly designed variable transformation allow bringing more variance into the leading principal components?

*Among the PCA input variables, especially $CO$, $CH_4$, and in the winter half year also $CO_2$ tend to have right-skewed PDFs. A logarithmic transformation, for example, would reduce the influence of the "long tails" of the PDFs on the principal component load-ings. However, a variable transformation would not completely remove the "long tails" because the PDFs rarely follow an idealized distribution such as a log-normal distribu-tion and they vary with the 2-month periods. Additionally, a variable transformation has the following side effect. If a logarithmic transformation is used for example and the original variable increases by a certain amount of units, then the transformed variable will not increase by a certain amount of units but by a certain factor. Although a vari-able transformation might somewhat increase the fraction of variance explained by the leading principal components in some 2-months, I have the impression that it would not significantly improve the classification of the entire data set.*

(2) The matching between air-mass regimes (I-IX) and air-mass classes (ML, UFT/SIN, HYBRID) is different in each two-month period (see Figure 8). The manuscript text

contains little or no information about the overarching logic. Why was this necessary? What criteria were used to attribute regimes to classes, how did these criteria change with the season?

In my opinion, the ad-hoc tuning of the method is a serious shortcoming. It is clearly a subjective component of the classification, and as such it cannot be exported to other sites. The authors do not explain this point in a satisfactory manner, and they probably should. Why wasn't it possible to design a fully objective classification rule? Formal methods to identify classification rules exist and could be used (see for instance chapter 14 in Wilks, 2011, Statistical Methods in the Atmospheric Sciences. DOI: 10.1016/B978-0-12-385022-5.00014-2).

*I realize that the mapping of air mass regimes to air mass classes is only shortly described in the manuscript. More details will be added. The loadings of the leading principal components changed with the 2-month periods (Fig. 6), which can be explained by seasonal changes in chemical processes (e.g. $CO_2$ emissions and uptake, photochemical $O_3$ production) and atmospheric dynamics (e.g. thermally induced uplift). Therefore, the characteristics of the air mass regimes (I-IX) changed with the 2-month periods and required a separate interpretation in each 2-month period. The air mass regimes were assigned to the air mass classes by visually comparing boxplots of the original input variables between the air mass regimes (Fig. 7). In the winter half year, the class ML was assigned if $CO$, $CH_4$, $CO_2$, $q$, and $\Delta\theta_v$ were relatively high and $p_{GAP}$ and $O_3$ were relatively low compared to the other regimes. The class UFT/SIN was assigned in the opposite case and the remaining regimes were assigned to the class HYBRID. Apart from $CO_2$ and $O_3$, the same criteria were used in the summer half year; $CO_2$ was required to be relatively low for the class ML and $O_3$ was not considered.*

*I agree that this subjective mapping of regimes to classes is a shortcoming. Nevertheless, it is based on typical qualitative differences between lifted and subsided air masses. The present study has the character of a pilot study to develop a novel mul-*

*tivariate approach for air mass classification. In future studies, a more objective and robust mapping of regimes to classes could be achieved by using a metric such as the median of a regime to define a threshold for "relatively high/low" characteristics and by using data from multiple years or more observatories.*

**References**

Forrer, J., Rüttimann, R., Schneiter, D., Fischer, A., Buchmann, B. and Hofer, P.: Variability of trace gases at the high-Alpine site Jungfraujoch caused by meteorological transport processes, Journal of Geophysical Research: Atmospheres, 105, 12241–12251, 2000.

Griffiths, A. D., Conen, F., Weingartner, E., Zimmermann, L., Chambers, S. D., Williams, A. G. and Steinbacher, M.: Surface-to-mountaintop transport characterised by radon observations at the Jungfraujoch, Atmospheric Chemistry and Physics, 14, 12763–12779, 2014.

Lugauer, M. and Winkler, P.: Thermal circulation in South Bavaria climatology and synoptic aspects, Meteorologische Zeitschrift, 14, 15–30, 2005.

Scheel, H., Sladkovic, R. and Kanter, H.: Ozone variations at the Zugspitze (2962 m asl) during 1996-1997, WIT Transactions on Ecology and the Environment, 35, 264–268, 10.2495/EURO990511, 1999.

Zellweger, C., Forrer, J., Hofer, P., Nyeki, S., Schwarzenbach, B., Weingartner, E., Ammann, M. and Baltensperger, U.: Partitioning of reactive nitrogen ($NO_y$) and dependence on meteorological conditions in the lower free troposphere, Atmospheric Chemistry and Physics, 3, 779–796, 2003.

---

## Referee Comment (RC2) · Anonymous Referee #1 · 21 May 2019

**Review of "Multivariate statistical air mass discrimination for the high-alpine observatory at the Zugspitze mountain, Germany" by Sigmund et al. 2019 (acp-2019-211)**
**submitted to "Atmospheric Chemistry and Physics" "**

**May 21, 2019**

In this study, the authors use a statistical approach to distinguish between different air masses at the high-alpine site Schneefernerhaus at Mt. Zugspitze in Germany. Based on different gas-phase and meteorological variables they performed a principal component analysis and define 9 air mass regimes. They validate the statistical approach with a mechanistic approach using the ceilometer-based mixing layer height and outline how this approach can be used for a real-time discrimination of air masses.

This study addresses a relevant topic in mountain meteorology and climate monitoring and falls into the scope of ACP. As far as I know, the method is novel and depicts a promising approach. It is clearly described and may thus be transferred to other high-alpine sites. Overall, the manuscript is well written and the outline is clear. Nevertheless, I have several detailed comments and suggestion which are given below and which should be considered by the authors before the manuscript is accepted for publication in ACP.

**1 Specific comments**

1. Introduction: The authors list studies done for many different high-altitude sites. However, one observatory which is not mentioned at all is Pic du Midi in the French Pyrenees. There was a recent study by Hulin et al. (2019) on atmospheric composition and the detection of thermally driven circulations with different methods, which should be cited as well.

2. p. 4, l. 22: "available at 1 min intervals". Are the values averaged over 1 min intervals or instantaneous?

3. p. 5, l. 2: What is the temporal resolution of the meteorological data?

4. p. 5, l. 3-5: How are the aerosol layer heights detected? With the manufacturer software or with a algorithm developed by the authors?

5. p. 5, l. 4: ">1": What is the unit? dB, B,...?

6. p. 5, l. 7: Is daylight saving time taken into account?

7. p. 5, l. 8: Why did the authors use a threshold of 66 % and not something else for the availability?

8. p. 6, l. 1ff: Why is there a time offset between the sites? Where the sites not synced to a time server? If the sites are not synced was there a shift of the time offset with time? E.g. what is the difference between Sept. 2013 and March 2014? How high is the correlation coefficient?

9. p. 6, l. 19: "exp" should not be italic.

10. p. 7, l. 16: Why only 89 % at the beginning? What happened to the remaining 11 %.

11. p. 8, l. 2: Why four classes? In the following lines only three air mass classes are defined?

12. p. 8, l. 24: What are "most suitable variables"? How are they defined? What are the criteria?

13. p. 9, l. 17: The figures should be referred to in the correct order, i.e. 8 not before 4.

14. p. 9, l. 19: What are the remaining measurements?

15. p. 9, l. 25: I was surprised to read about a marine boundary layer considering the location of Zugspitze. Where does the marine air mass come from? The authors speculate about that later in the manuscript, but I think a hint about its origin should already be given here.

16. p. 10, l. 8ff: What is the typical stability distribution? This could e.g. be checked with a histrogram.

17. p. 10, l. 12: GAP is a valley floor station. Why is this station used to detect strong synoptic forcing? Wouldn't it make more sense to use high-elevation sites?

18. p. 10, l. 16: Why does condition a) requires anabatic wind OR UFS below $MLH_{GAP}$? Why not AND?

19. p.10, l. 30-31: The authors state that PC1 was always a meaningful indicator while PC2 did not always allow for an unambiguous interpretation. On what is that assumption based?

20. Sect. 3.1: I found this part of the result section hard to follow and it might be difficult for readers not familiar with PCA to understand the interpretation of the results. It might be helpful to give a detailed example at the beginning on how to read and understand the loading diagrams in Fig. 6. What does a low score mean? Low absolute values or large negative values? In the text (p. 11, l. 1-6), the authors talk about scores while in Fig. 6 loadings are

shown. How does this relate?

21. p. 11, l. 32: "... air masses (Fig. 7a-g): compared ..." To make clear that the explanation why it was consistent is following.

22. p. 14, l. 15: This additional criteria of no clouds below 4 km, should be moved to Sect 2.5.

23. p. 14, l. 21: " .. this winds were not thermally induced BUT the $MLH_{GAP}$ suggested non-ML air masses." BUT does not make sense. It should be AND.

24. p. 14, l. 28ff: What about LRMD and MBL/UFT air mass classes?

25. p. 15, l. 27: What differed between the six 2-month periods?

26. Fig. 7: Why not stick to the numbers I-IX for the regimes instead of introducing the long names for the regimes. This would make it more comparable with Figs. 3 and 8. The colours for the air mass classes should be brighter like in Fig. 8 and 9. Adjust the scales of the subplots to make them clearer to read (e.g. Figs. 7o, 7p, 7q, 7r).

27. Fig. 8: I probably understand it wrong, but how can regime VI belong to ML? In Fig. 3 there is no connector between ML and regime IV. Also, how can I and II belong to HYBRID? It would be good to enhance the boxes of the label to make the hatched areas better visible.

28. Fig. 9: Maybe add "UFS below MLH..." or "UFS above..." to the label and maybe refer to the text (p. 10, l. 14ff) where the 3 conditions are explained.

**References**

Hulin, M., Gheusi, F., Lothon, M., Pont, V., Lohou, F., Ramonet, M., Delmotte, M., Derrien, S., Athier, G., Meyerfeld, Y., Bezombes, Y., Augustin, P., and Ravetta, F.: Observations of Thermally Driven Circulations in the Pyrenees: Comparison of Detection Methods and Impact on Atmospheric Composition Measured at a Mountaintop, J. Appl. Meteor. Climatol., 58, 717–740, doi:10.1175/JAMC-D-17-0268.1, 2019.

---

## Author Comment (AC1) · 3 Aug 2019

The final response and marked-up manuscript versions of the main article and its supplement are attached in the form of a supplement.

Please also note the supplement to this comment:
https://www.atmos-chem-phys-discuss.net/acp-2019-211/acp-2019-211-AC1-supplement.zip

---

## Author Response (AR1)

**Final response**

Our responses are marked in *italic and blue* and were directly inserted below each comment of the referees. Changes in the manuscript are described below each comment and are highlighted in the attached version of the paper.

We thank both referees for their helpful comments and the time they spent with the review.

**5 Comments from Referee #1**

1. Introduction: The authors list studies done for many different high-altitude sites. However, one observatory which is not mentioned at all is Pic du Midi in the French Pyrenees. There was a recent study by Hulin et al. (2019) on atmospheric composition and the detection of thermally driven circulations with different methods, which should be cited as well.

Thanks for pointing out that study. In the new manuscript version, the study is cited in the introduction.

2. p. 4, l. 22: "available at 1 min intervals". Are the values averaged over 1 min intervals or instantaneous?
 The values are averaged over 1 min intervals. In the new manuscript version, we use the phrase "available as 1 min averages".

3. p. 5, l. 2: What is the temporal resolution of the meteorological data?

The standard meteorological data was available as 10 min averages except at the Schneefernerhaus where 1 min averages 15 were available. This is now mentioned in the manuscript.

4. p. 5, l. 3-5: How are the aerosol layer heights detected? With the manufacturer software or with a algorithm developed by the authors?

The manufacturer software was used, which is now mentioned in the manuscript.

5. p. 5, l. 4: ">1": What is the unit? dB, B,...?

30

20 This statement was taken from (Heese et al., 2010) who expressed the signal-to-noise ratio in a dimensionless number.

6. p. 5, l. 7: Is daylight saving time taken into account?

We checked whether the time stamp referred to daylight saving time during the summer half year but this was not the case. However, some measurements were recorded with UTC time, which was converted to local standard time.

7. p. 5, l. 8: Why did the authors use a threshold of 66 % and not something else for the availability?

25 We agree that the minimum data availability of 66 % per interval is somewhat arbitrary. However, it guarantees that the averaged values are representative for most of the time interval and not only for a small part of the interval in the case of data gaps.

8. p. 6, l. 1ff: Why is there a time offset between the sites? Where the sites not synced to a time server? If the sites are not synced was there a shift of the time offset with time? E.g. what is the difference between Sept. 2013 and March 2014? How high is the correlation coefficient?

After contacting the manufacturer of the weather stations, we realized that the time stamps are synced to a time server and should not differ much between the sites. The initially supposed time offsets based on crosscorrelation may be due to an inexact horizontal alignment of the pyranometers. Therefore, we repeated the analyses without shifting the time and updated the text and the figures. This had a negligible effect on  $\Delta \theta_v$  and the statistical air mass classification because initially, a time offset had

35 only be supposed for the sites ZPLT, Schachen, Kreuzalm, and Felsenkanzel and not for GAP, Brandwiese, and ZSG. Omitting the time shift, had a small effect on the mechanistic classification and slightly improved the agreement between the mechanistic and the statistical classifications.

9. p. 6, l. 19: "exp" should not be italic.

Thanks, this is now corrected.

20

10. p. 7, l. 16: Why only 89 % at the beginning? What happened to the remaining 11 %. During 11 % of the time, the ceilometer could not determine any aerosol layer height, most likely due to fog or precipitation.

5 11. p. 8, l. 2: Why four classes? In the following lines only three air mass classes are defined? *"Four" was a mistake, it should be three. We have corrected this now.*

12. p. 8, l. 24: What are "most suitable variables"? How are they defined? What are the criteria?

To be more precise, we changed the phrase to "variables with an expected unambiguous link to vertical transport processes and a high data availability". These variables include CO, CH4, CO2, O3, specific humidity, air pressure at GarmischPartenkirchen, and Δθv and are now mentioned directly afterwards in the manuscript.

13. p. 9, l. 17: The figures should be referred to in the correct order, i.e. 8 not before 4. *Figure 8b is now Fig. 3b so that the figures are referred to in the correct order.*

**14. p. 9, l. 19: What are the remaining measurements?**

The remaining measurements refer to the gases NOy, NOx, 222Rn, 7Be, HCHO, 222Rn, the aerosol quantities N90, eBC,
PM10, and the standard meteorological variables precipitation, relative humidity, temperature, global radiation. Because these measurements are already mentioned in Sect. 2.2, the new manuscript version summarizes them as "remaining chemical (e.g. NOy, NOx, 222Rn) and standard meteorological measurements (e.g. precipitation, relative humidity)".

15. p. 9, l. 25: I was surprised to read about a marine boundary layer considering the location of Zugspitze. Where does the marine air mass come from? The authors speculate about that later in the manuscript, but I think a hint about its origin should already be given here.

We adopt this suggestion and now mention the Atlantic Ocean as a potential origin of air masses in that part of the manuscript.

16. p. 10, l. 8ff: What is the typical stability distribution? This could e.g. be checked with a histrogram.

- We address this question by adding a short section in the supplement, including histograms of  $\Delta \theta_v$  for the periods June–July and December–January (Fig. S1a). Note that we have additionally changed the sign convention for  $\Delta \theta_v$  (see comment 13 of Referee #2).  $\Delta \theta_v$  was almost always positive, indicating stable conditions. In June and July,  $\Delta \theta_v$  ranged between -5 K and +21 K whereas in December and January, it was generally more positive with values between 4 K and 31 K. Additionally, we included the histogram, from which the threshold of  $\Delta \theta_v = 8$  K between anabatic and katabatic winds was determined (Fig. S1b).
- 30 17. p. 10, l. 12: GAP is a valley floor station. Why is this station used to detect strong synoptic forcing? Wouldn't it make more sense to use high-elevation sites?

We restricted the wind velocities at ZPLT and GAP to  $

Figure R1. Histograms of the horizontal wind velocities  $(v_h)$  at GAP and ZPLT for potential katabatic winds, i.e. for downvalley wind direction sectors at both stations and strongly stable conditions ( $\Delta \theta_v \ge 8$  K). The dashed line shows the threshold of 3 m s-1 for katabatic winds.

18. p. 10, l. 16: Why does condition a) requires anabatic wind OR UFS below MLHGAP? Why not AND?

The mixing layer height can be heterogeneous, especially in mountainous terrain.  $MLH_{GAP}$  is measured in the center of the valley atmosphere. Due to thermally driven upslope winds, boundary layer air can reach higher altitudes at the mountain slopes compared to the valley center (Henne et al., 2004; Gohm et al., 2009). Thus, the UFS could be influenced by the boundary layer while  $MLH_{GAP}$  is below the UFS level. Additionally, a residual layer could result in boundary layer air masses at the

5 layer while MLHGAP is below the UFS level. Additionally, a residual layer could result in boundary layer air masses at the UFS during night while anabatic winds are absent.

19. p.10, l. 30-31: The authors state that PC1 was always a meaningful indicator while PC2 did not always allow for an unambiguous interpretation. On what is that assumption based?

This statement is explained in the following paragraph in which the PC loadings are discussed. To make that clear, we inserted 10 the phrase ", which will be explained in the following".

20. Sect. 3.1: I found this part of the result section hard to follow and it might be difficult for readers not familiar with PCA to understand the interpretation of the results. It might be helpful to give a detailed example at the beginning on how to read and understand the loading diagrams in Fig. 6. What does a low score mean? Low absolute values or large negative values? In the text (p. 11, l. 1-6), the authors talk about scores while in Fig. 6 loadings are shown. How does this relate?

15 With low score, we mean large negative linear combinations according to Eq. 5. In the new manuscript, we explain that variables with a high absolute loading largely determine the PC scores and give an example on how loadings affect PC scores. If an original variable is much higher than its 2-month mean value and its loading on PC1 is strongly negative then the variable will strongly contribute to a large negative score of PC1.

21. p. 11, l. 32: "... air masses (Fig. 7a-g): compared ..." To make clear that the explanation why it was consistent is following. *We adopted the suggestion of inserting a colon.*

22. p. 14, l. 15: This additional criteria of no clouds below 4 km, should be moved to Sect 2.5. *As suggested, we moved this criterion to Sect. 2.5, in which the mechanistic approach is described.*

20

23. p. 14, l. 21: " .. this winds were not thermally induced BUT the  $MLH_{GAP}$  suggested non-ML air masses." BUT does not make sense. It should be AND.

We agree and replace BUT by AND. The same applies to two similar sentences in the same section and in Sect. 2.5.

24. p. 14, l. 28ff: What about LRMD and MBL/UFT air mass classes?

5 LRMD and MBL/UFT are subclasses of HYBRID. To make that clear, we write "HYBRID including LRMD and MBL/UFT" in the new manuscript version.

25. p. 15, l. 27: What differed between the six 2-month periods?

The first two principal components, their interpretation, and thus, the mapping of air mass regimes to air mass classes differed between the 2-month periods. We changed the sentence into the following one: "but the principal components and their interpretation differed between the six 2-month periods".

10 *interpretation differed between the six 2-month periods*".

26. Fig. 7: Why not stick to the numbers I-IX for the regimes instead of introducing the long names for the regimes. This would make it more comparable with Figs. 3 and 8. The colours for the air mass classes should be brighter like in Fig. 8 and 9. Adjust the scales of the subplots to make them clearer to read (e.g. Figs. 7o, 7p, 7q, 7r).

To make Fig. 7 more comparable with Fig. 3 and 8, we replaced the long regime names by the numbers I to IX. The same was done for the figures in the supplement. Also, the colours and scales in Fig. 7 were adjusted as suggested.

27. Fig. 8: I probably understand it wrong, but how can regime VI belong to ML? In Fig. 3 there is no connector between ML and regime IV. Also, how can I and II belong to HYBRID? It would be good to enhance the boxes of the label to make the hatched areas better visible.

In Fig. 3, the mapping between of air mass regimes to air mass classes is only shown for February and March. In other 2-month
periods, the mapping was different, which is now explained in the figure caption. In Fig. 8, the legend is now bigger and the hatched areas are better visible.

28. Fig. 9: Maybe add "UFS below MLH..." or "UFS above..." to the label and maybe refer to the text (p. 10, l. 14ff) where the 3 conditions are explained.

We adopted these suggestions.

**25 Comments from Referee #2**

(1) Pre-processing of the data before use in the statistical classification method is limited to standardization (that is, adjustment of the sample mean to 0 and of the sample variance to 1). I am wondering whether any slightly more sophisticated pre-processing could be beneficial.

For instance:

30 (a) Some of the variables in the data matrix have well-defined seasonal and diurnal cycles. Would it be possible to determine average annual and daily cycles, and to remove them from the data set? Performing the analysis on deviations from the average cycles might improve classification results.

In our analysis, the seasonal course was roughly removed by subtracting the 2-month mean in each of the 2-month periods when standardizing the data. We added a sentence in Sect. 2.4.2 to make that clear. In principle, it would be beneficial to

- 35 remove the seasonal course more accurately so that the leading principal components would only depend on the shorter-term variability of the time series. In the beginning of the data analysis, we considered the determination and removal of the seasonal course with a spectral approach such as a wavelet filter. However, this idea was abandoned because it requires knowledge of the time series within a window centered at the time of interest and is thus not applicable to real-time operational mode. It would be possible to remove an average seasonal cycle from each variable based on a larger, multi-year data set. Due to
- 40 interannual variability, however, this approach would only partly remove the seasonal course. For example, Scheel et al. (1999)

showed that the monthly mean  $O_3$  mixing ratios at Mt. Zugspitze can strongly differ between individual years and the 10-year ensemble. Therefore and because a large fraction of the seasonal variability is already removed by subtracting the 2-month mean values, we doubt that the removal of an average seasonal cycle would substantially improve the classification results.

- The mean diurnal cycles of the analyzed variables reflect shifts between air masses that we aim to identify. Afternoon maxima of atmospheric constituents such as water vapor, CO, 222Rn, and NOy at high-alpine sites have been explained by thermally induced uplift processes including anabatic winds (Forrer et al., 2000; Zellweger et al., 2003; Griffiths et al., 2014). These conditions are typically accompanied by afternoon minima of atmospheric stability and air pressure. The latter is associated with plain to mountain winds in the Northern Alps (Lugauer and Winkler, 2005). Removing the average diurnal cycle would complicate the classification of air masses in the case of thermally induced vertical transport.
- 10 (b) PCA does not require the data to follow multivariate normal distributions, but its results can often be interpreted more easily if they do. It strikes me that most variables are concentrations, therefore their PDF will certainly be markedly non-Gaussian. Would a cleverly designed variable transformation allow bringing more variance into the leading principal components?

Among the PCA input variables, especially CO,  $CH_4$ , and in the winter half year also  $CO_2$  tend to have right-skewed PDFs. A logarithmic transformation, for example, would reduce the influence of the "long tails" of the PDFs on the principal component

- 15 loadings. However, a variable transformation would not completely remove the "long tails" because the PDFs rarely follow an idealized distribution such as a log-normal distribution and they vary with the 2-month periods. Additionally, a variable transformation has the following side effect. If a logarithmic transformation is used for example and the original variable increases by a certain amount of units, then the transformed variable will not increase by a certain amount of units but by a certain factor. Although a variable transformation might somewhat increase the fraction of variance explained by the leading
- 20 principal components in some 2-month periods, we think that it would not significantly improve the classification of the entire data set.

(2) The matching between air-mass regimes (I-IX) and air-mass classes (ML, UFT/SIN, HYBRID) is different in each twomonth period (see Figure 8). The manuscript text contains little or no information about the overarching logic. Why was this necessary? What criteria were used to attribute regimes to classes, how did these criteria change with the season?

- In my opinion, the ad-hoc tuning of the method is a serious shortcoming. It is clearly a subjective component of the classification, and as such it cannot be exported to other sites. The authors do not explain this point in a satisfactory manner, and they probably should. Why wasn't it possible to design a fully objective classification rule? Formal methods to identify classification rules exist and could be used (see for instance chapter 14 in Wilks, 2011, Statistical Methods in the Atmospheric Sciences. DOI: 10.1016/B978-0-12-385022-5.00014-2).
- 30 We realize that the mapping of air mass regimes to air mass classes was only shortly described in the manuscript. Now, more details are included in Sect. 2.4.3. The loadings of the leading principal components changed with the 2-month periods (Fig. 6), which can be explained by seasonal changes in chemical processes (e.g. CO2 emissions and uptake, photochemical O3 production) and atmospheric dynamics (e.g. thermally induced uplift). Therefore, the characteristics of the air mass regimes (I-IX) changed with the 2-month periods and required a separate interpretation in each 2-month period. The air mass regimes
- 35 were assigned to the air mass classes by visually comparing boxplots of the original input variables between the air mass regimes (Fig. 7). In the winter half year, the class ML (now renamed as BL, see comment 4 of Referee #2) was assigned if CO, CH4, CO2, and q were relatively high and  $\Delta \theta_v$ ,  $p_{GAP}$  and O3 were relatively low compared to the other regimes; note that the sign of  $\Delta \theta_v$  was changed so that low values indicate a low static stability (see comment 13 of Referee #2). The class UFT/SIN was assigned in the opposite case and the remaining regimes were assigned to the class HYBRID. Apart from CO2 and O3,
- 40 the same criteria were used in the summer half year; CO2 was required to be relatively low for the class BL and O3 was not considered.

We agree that this subjective mapping of regimes to classes is a shortcoming. Nevertheless, it is based on typical qualitative differences between lifted and subsided air masses. The present study has the character of a pilot study to develop a novel multivariate approach for air mass classification. In future studies, a more objective and robust mapping of regimes to classes

45 could be achieved by using a metric such as the difference between the median of a regime and the overall median to define a threshold for "relatively high/low" characteristics and by using data from multiple years or more observatories. The supervised

**classification methods, which are described in Chapter 14 in Wilks (2011), could only be applied to our problem if a reliable test data set with known air mass classes was available and included a variety of meteorological conditions.**

(3) Note: the line numbering in pages 2-end is wrong, i.e., the 6th line from the top is labelled as "5" and so on, as if the the first line were 0. In what follows, I'm using this unusual "zero-based" system.

5 This issue has been corrected.

(4) Nomenclature. The first air-mass class is labelled ML, for "mixing layer". I'm wondering if this is appropriate. A mixed layer, by definition, has nearly adiabatic lapse rate. The boundary layer (BL) is not always well-mixed, especially at night. On page 1, line 12, it is stated that "the terms ML and BL can be defined synonymously ...". In my mind, the two concepts are quite distinct. I'd rather say that the BL might sometimes include a ML. I don't really have a strong opinion on this

10 matter, but anyway I suggest renaming the first air-mass class to BL, for "boundary layer". A similar comment applies to "mixing layer height" (MLH). This should probably become "boundary layer height" (BLH), in particular because, according to the description of the wavelet detection algorithm, MLH/BLH potentially includes multiple aerosol layers. These typically develop in connection with inversion layers, i.e., non-mixed parts of the atmosphere.

We agree that the definitions of the boundary and mixing layers are based on different concepts. The boundary layer (BL) is affected by surface forcings such as the turbulent transfer of momentum, heat, and matter while the mixing layer specifically refers to the dispersion of surface-emitted atmospheric constituents (Stull, 1988; Seibert et al., 2000). In the new manuscript version, we cite definitions of the boundary and mixing layers. We point out that we consider a residual layer and elevated aerosol layers, which were influenced by the surface within a time scale of one diurnal cycle, as parts of the boundary and mixing layers, as suggested by Reuten et al. (2007). Because the term mixing layer is usually restricted to the well-mixed layer

20 adjacent to the surface, we renamed the air mass class ML as BL. However, we keep the term mixing layer height (MLH) because the ceilometer measures the aerosol backscatter profile, which reflects the dispersion of surface-emitted particles.

(5) Page 1, title. "Discrimination" or "classification"? The two terms have slightly different meanings. See again chapter 14 in Wilks 2011.

We now use the word "classification" instead of "discrimination" because discrimination would be based on training data,

25 for which the groups (air mass classes) are already known, while classification refers to the attribution of test data to groups (Wilks, 2011).

(6) Page 1, lines 8 and 12. Use of the word "classifiable". I believe these statements should be formulated more clearly. As they are now, they seem to allude to the intrinsic "ability" of the methods to separate the events, and seem to suggest that the statistical method permits to obtain a meaningful classification much more often than the mechanistic one (78 % of the time as opposed to 25 %). Instead, these two percentage only represent the availability of the input data for the two methods. Please

30 opposed to 25 %). Instead, these two percentage only represent the availability of the input data for use something like: "Due to data gaps, only x % of the investigated year could be classified".

To avoid misunderstandings, we avoid the expression "classifiable cases" and now use the phrases "... input data was available in 78 % ... " and "Due to data gaps, only 25 % of the cases could be classified ... ". In the rest of the paper, we use "classified cases" instead of "classifiable cases".

- 35 (7) Page 2, lines 10-11. Foehn flows are listed among processes that cause air mass lifting. This is inexact and quite confusing. Foehn is a fall wind: its dynamics are inextricably tied to air-mass descent (not lifting!) on the lee side of a mountain range. That said, intense foehn certainly causes mechanical mixing of the lower atmosphere, which may result in transport of chemical species from the PBL to the free troposphere. Zellweger et al (2003, cited in the manuscript) list foehn among the meteorological conditions in which free-troposphere air masses are mixed with PBL air masses (page 781, top of second column).
- 40 Correctly, Zellweger et al (2003) do not mention "lifting" in relation to foehn. Please revise. The comment also applies to page 2, line 23 and to page 12, line 15.

In the case of south foehn, Mt. Zugspitze is located on the lee side of the Alps where the air flow descends. Nevertheless, foehn events can be associated with lifting on the windward side of the mountain range and thus transport air masses from the BL to

high-alpine sites. If the southern Alps experience north foehn, Mt. Zugspitze is located on the windward side of the Alps where the air flow typically ascends and lifts BL air masses. In the new manuscript version, we list foehn among transport processes that cause air mass lifting or mixing. We also mention that foehn winds descend on the lee side of a mountain range and can be associated with air mass lifting on the windward side. On page 12, we now speak of an "influence of BL air masses" with respect to foehn instead of an "uplift of BL air masses".

(8) Page 2, line 30. "... because the MLH is a meteorological quantity". Wording could be more careful here. MLH/BLH is not a directly measured quantity, but rather an estimate obtained from measurements of other quantities. The determination of MLH/BLH from a vertical profile can be quite tricky, too. I'd rather say something like: "... because determination of the MLH from vertical profiles of measured quantities requires a-priori meteorological knowledge".

10 We adopted this suggestion.

5

(9) Page 5, line 17. "... set zero"  $\rightarrow$  "... set to zero". *Done.*

(10) Page 5, line 21. Please delete the blank space before the full stop. *Done.*

15 (11) Page 6, line 17. "... using the Clausius Clapeyron equation". Or rather a numerical approximation? There are many such formulas: Goff-Gratch, Magnus-Tetens, Bolton ... which one?

We used the following form of the Clausius Clapeyron equation (e.g. Wallace and Hobbs, 2006),

$$\frac{1}{e_s}\frac{de_s}{dT} = \frac{L_v}{R_v T^2},\tag{R1}$$

where  $e_s$  (hPa) is saturation vapor pressure, T (K) is temperature,  $L_v$  (J kg-1) is latent heat of evaporation, and  $R_v$  (J kg-1 K-1) 20 is specific gas constant for water vapor. Integration of Eq. R1 yields

$$e_s(T) = e_s(T_0) \exp\left\{\frac{L_v}{R_v} \left(\frac{1}{T_0} - \frac{1}{T}\right)\right\},\tag{R2}$$

where  $T_0$  (K) is a reference temperature with known  $e_s$ .

(12) Page 6, line 19. I think the standard notation should be either  $e^x$  or  $\exp\{x\}$ . Also, please replace  $T_v$  by  $\overline{T}_v$ , to indicate the vertical averaging.

25 *Done*.

(13) Page 7, line 2. I personally find the sign convention counterintuitive. Although static stability conventionally corresponds to  $d\theta/dz > 0$ , here  $\Delta\theta$  is greater than zero when  $\theta$  decreases with height. Why not computing  $d\theta/dz$ ?

Noting that the sign convention for  $\Delta \theta_v$  was counterintuitive, we changed this sign convention. Now, positive values indicate stable conditions. This change corresponds to multiplying  $\Delta \theta_v$  by -1. Consequently, the sign of the PC loadings of  $\Delta \theta_v$  changed as well (Fig. 6). Dividing  $\Delta \theta_v$  by  $\Delta r$  would be an elternative but would not significantly change the results because

30 changed as well (Fig. 6). Dividing  $\Delta \theta_v$  by  $\Delta z$  would be an alternative but would not significantly change the results because  $\Delta z$  does not vary much.

(14) Page 7, lines 20-21. "The MLH attribution was based on the idea that the MLH varies only gradually". I am not sure this is always appropriate over mountains. Horizontal advection of aerosol layers due to mountain venting can cause spatial and temporal discontinuities in MLH.

35 We notice this shortcoming and now mention it in the manuscript. For most of the time, however, we expect a gradual variation of the MLH.

(15) Page 10, lines 9-10. Why using -8 K to discriminate "weak" and "strong" static stability?

The threshold of -8 K (now +8 K because of the changed sign convention, see comment 13 of Referee #2) was determined by comparing the histograms of  $\Delta \theta_v$  for winds coming from the upvalley or downvalley wind direction sectors at GAP and ZPLT. The histograms are now included in the supplement as Fig. S1b. Using the intersect of the histograms corresponds to minimizing the number of data points that are excluded from potential anabatic and katabatic winds.

(16) Page 11, line 29. Are  $CH_4$  and  $CO_2$  pollutants?

5

Originally, we considered  $CH_4$  and  $CO_2$  as pollutants. However, the paragraph under consideration was removed because the long names of the air mass regimes were replaced by roman numbers. See comment 26 of Referee #1.

Please find below the marked-up manuscript versions of the main article and its supplement.

2Bavarian Environment Agency, Augsburg, Germany

[revised manuscript text omitted]

---

## Referee Report (RR1)

**Multivariate statistical air mass classification for the high-alpine observatory at the Zugspitze mountain, Germany**

My two most important comments on the first version of this manuscript were:

(1) Better classification results could have been obtained by adequately pre-processing the observations (e.g., removing seasonal and daily cycles, or transforming variables to achieve approximately Gaussian PDFs).
(2) The matching between air-mass regimes and air-mass classes varies seasonally and is based on subjective considerations, limiting the applicability of the method to other sites.

Generally, the authors used reasonable arguments to defend their work. In particular:

(1) Seasonal variability cannot be filtered adequately without a very long dataset at hand, especially for some quantities with marked interannual variability. Diurnal variability must be retained in the classification design, because it is important in order to identify BL air masses.
(2) The seasonally-dependent correspondence between air-mass regimes and air-mass classes is justified by the fact that the characteristic properties of air masses in terms of the PCA input variables change seasonally (e.g., BL air has relatively high/low $CO_2$ concentration compared to other air masses, respectively in winter/summer). While unclear in the previous version of the manuscript, this aspect is now appropriately explained.

Two other recommendations of mine were: i) To consider transforming variables before the PCA; ii) To work towards a fully objective classification method, i.e., not based on subjective analysis of air-mass regimes. The authors opted for not following these suggestions at the moment, and I find it understandable. Anyway, I encourage them to explore these issues in future work.
I have two final minor comments:

(1) Concerning my previous comment 7: I appreciate the improved presentation of foehn-related issues, nevertheless I suggest to rewrite lines 11-12 this way: "Foehn winds descend on the lee side of a mountain range BUT can be associated with air mass lifting on the windward side, AND WITH ENHANCED TURBULENT MIXING IN THE LEE".
(2) Concerning my previous comment 11: Equations R1 and R2 in the author response are indeed an approximation of the Clausis-Clapeyron equation, This formulation assumes that the specific volume of liquid water is negligible and that the latent heat of vaporization does not depend on temperature. It would be possible to get a better estimate of the vapor pressure in a larger temperature range using different approximations. Anyway, the errors are expected to be small and justifiable for the purposes of this article.

---

## Author Response (AR2)

**Author Response**

**Our responses are marked in *italic and blue* and were directly inserted below each comment. Changes in the manuscript are listed in Sect. 2 of this response and are highlighted in the attached version of the manuscript (Sect. 3).**

*We thank Referee #2 for reviewing our revised manuscript. The valuable comments help to further improve the paper.*

**1 Final comments from Referee #2**

My two most important comments on the first version of this manuscript were:

(1) Better classification results could have been obtained by adequately pre-processing the observations (e.g., removing seasonal and daily cycles, or transforming variables to achieve approximately Gaussian PDFs).

(2) The matching between air-mass regimes and air-mass classes varies seasonally and is based on subjective considerations, limiting the applicability of the method to other sites.

Generally, the authors used reasonable arguments to defend their work. In particular:

(1) Seasonal variability cannot be filtered adequately without a very long dataset at hand, especially for some quantities with marked interannual variability. Diurnal variability must be retained in the classification design, because it is important in order to identify BL air masses.

(2) The seasonally-dependent correspondence between air-mass regimes and air-mass classes is justified by the fact that the characteristic properties of air masses in terms of the PCA input variables change seasonally (e.g., BL air has relatively high/low CO2 concentration compared to other air masses, respectively in winter/summer). While unclear in the previous version of the manuscript, this aspect is now appropriately explained.

Two other recommendations of mine were: i) To consider transforming variables before the PCA; ii) To work towards a fully objective classification method, i.e., not based on subjective analysis of air-mass regimes. The authors opted for not following these suggestions at the moment, and I find it understandable. Anyway, I encourage them to explore these issues in future work.

*We appreciate these suggestions and will consider them in future projects.*

I have two final minor comments:

(1) Concerning my previous comment 7: I appreciate the improved presentation of foehn-related issues, nevertheless I suggest to rewrite lines 11-12 this way: "Foehn winds descend on the lee side of a mountain range BUT can be associated with air mass lifting on the windward side, AND WITH ENHANCED TURBULENT MIXING IN THE LEE".

*We adopted this suggestion, which describes all relevant transport mechanisms related to foehn in a precise and easy-to-follow way.*

(2) Concerning my previous comment 11: Equations R1 and R2 in the author response are indeed an approximation of the Clausis-Clapeyron equation, This formulation assumes that the specific volume of liquid water is negligible and that the latent heat of vaporization does not depend on temperature. It would be possible to get a better estimate of the vapor pressure in a larger temperature range using different approximations. Anyway, the errors are expected to be small and justifiable for the purposes of this article.

*We agree that Eq. (R1) and (R2) in the previous author response follow from the Clausis-Clapeyron equation if the two mentioned approximations are made, namely: (i) The specific volume of liquid water is negligible compared to that of water vapor and (ii) the latent heat of vaporization is independent of temperature. These approximations are common in meteorology and we expect them to cause negligible errors in regards to the purpose of this study.*

**2  List of changes in the manuscript**

The following page and line numbers refer to the marked-up manuscript version.

- title: The word "Mountain" was capitalized because, according to the journal's house standard, generic geographic terms are capitalized when they are part of a place name ("Zugspitze Mountain"). The same applies to the expression "Zugspitze Summit" on p. 4, l. 11.

- p. 2, l. 12: The description of foehn flows is restated.

- p. 6, l. 27ff: We include the approximated and integrated form of the Clausius-Clapeyron equation and mention the approximations involved. A textbook reference is given.

- p. 10, l. 1 and p. 11, l. 32: The number of the referenced equation is now surrounded by parantheses.

- p. 19f: The DOI is now specified correctly and consistently throughout the references.

- p. 3, l. 3 of the supplement: A DOI was added to the reference.

**3  Marked-up manuscript version**

[revised manuscript text omitted]
 whereas in December and January, it was generally more positive with values between $4$ K and $31$ K (Fig. S1a). Upvalley winds at both GAP and ZPLT were generally associated with a

5   less negative $\Delta\theta_v$ compared to downvalley winds, which gave rise to the threshold of $8$ K for the distinction between anabatic and katabatic winds (Fig. S1b).

**S2 Wind patterns at Zugspitze summit in February and March**

Figure S2 shows the wind patterns at ZSG for each of the nine air mass regimes in the 2-month period February–March. The regimes I , II, and III indicated that BL air masses were often associated with southeasterly to southerly winds with varying

10   velocities (Fig. S2a-c). Some of these air masses may reflect south foehn events, especially in the case of strong southerly winds.

For all other regimes, the wind direction was more variable. Strong southerly winds were also included in the regimes IV, V, and VII (Fig. S2d,e,g), which suggests that foehn flows are not always associated with a strong uplift on the windward side of the Alps (Seibert, 1990) and can result in varying air mass characteristics. Regime VII exhibited the highest mean wind

15   velocity ($10.22$ m s$^{-1}$) among the regimes, which would be in line with a fast transport of the air masses from the marine boundary layer to the UFS (Fig. S2g).

The regimes VIII and IX, which were dominated by UFT/SIN air masses, exhibited similar mean wind velocities compared to the regimes of BL air masses (Fig. S2).

**S3 Statistical classification: Case study in March**

20   In a case study, the regimes and classes of air masses were highlighted in the measured time series to gain insight in the transport processes involved and to check the plausibility of the classification. The period from 1 to 13 March 2014 mainly included two phases with contrasting air mass characteristics (Fig. S3). From 1 to 6 March, the three air mass regimes, which were attributed to the class BL, dominated (Fig. S3a) – mainly due to high CO (Fig. S3b), $CH_4$ (Fig. S3c), and $CO_2$ mixing ratios (Fig. S3d) that peaked on 5 and 6 March. From 7 to 12 March, the two air mass regimes, which were attributed to the

class UFT/SIN in the absence of LRMD, dominated – due to low $CO$, $CH_4$, and $CO_2$ mixing ratios, predominantly low $q$ (Fig. S3e), and high $O_3$ mixing ratios (Fig. S3f). On 13 March, regime VII indicated ambiguous air masses that originated either from the lower free troposphere or the marine boundary layer (Fig. S3a).

$\Delta\theta_v$ showed weak and strong diurnal variations in the phases from 1 to 6 March and from 7 to 13 March, respectively (Fig. S3g), indicating a shift from cloudy to clear-sky conditions, as confirmed by $R_g$ measurements (Fig. S3s). $p_{\mathrm{GAP}}$ reached a minimum on 3 March, increased continuously and strongly until 6 March, and remained high until 13 March (Fig. S3h). These observations suggest that the BL air masses were lifted by a low pressure system and associated fronts whereas the UFT/SIN air masses descended in a high pressure system. This interpretation was supported by high rH (Fig. S3p), precipitation (Fig. S3r), and a low $R_g$ (Fig. S3s) during the strong pressure increase and low rH, absent precipitation and high $R_g$ during the high pressure phase.

The remaining chemical measurements were in line with the classification. $NO_y$, $NO_x$ (Fig. S3m), and $^{222}$Rn concentrations (Fig. S3n) were substantially higher and $^7$Be concentrations (Fig. S3o) were much lower for the BL than for the UFT/SIN air masses in the case study. The eBC (Fig. S3j) and $N_{90}$ concentrations (Fig. S3k) tended to be higher for the BL than for the UFT/SIN air masses but temporary wet deposition resulted in strong variations (Fig. S3j,k). The $PM_{10}$ concentration was only slightly higher for the BL air masses (median of 3.0 $\mu$g m$^{-3}$) than for the UFT/SIN air masses (median of 2.3 $\mu$g m$^{-3}$) (Fig. S3i). On 11 March 2014, a stratospheric intrusion was evident from exceptionally high Be-7 (Fig. S3o) and $O_3$ concentrations (Fig. S3f) of 26 mBq m$^{-3}$ and 79 ppb, respectively, and a low relative humidity of 12 % (Fig. S3p).

The ceilometer-based MLH at GAP was mostly missing in the period from 1 to 6 March because the uplift of BL air masses was associated with low-level clouds and precipitation (Fig. S3v). From 7 to 12 March, the MLH at GAP mostly showed diurnal variations with afternoon maxima but remained at least 300 m lower than the level of the UFS, which is in line with the statistical classification. Wind direction (Fig. S3t) and velocity (Fig. S3u) at ZSG did not differ significantly between BL and UFT/SIN air masses in the case study.

**References**

Seibert, P.: South Foehn Studies Since the ALPEX Experiment, Meteorology and Atmospheric Physics, 43, 91–103, https://doi.org/10.1007/BF01028112, 1990.

[Figure]

**Figure S1.** (a) Histograms of the range of the pseudo-vertical profile of virtual potential temperature ($\Delta\theta_v$) for (a) the periods from June to July and from December to January and for (b) upvalley and downvalley wind direction sectors at the stations GAP and ZPLT. Positive values represent stable conditions.

[Figure]

**Figure S2.** Windrose plots showing the frequency of counts by wind direction and velocity (v) at Zugspitze summit for the nine air mass regimes (I to IX) in February and March 2014. PC1 and PC2 denote the first and second principal components, respectively.

[Figure]

**Figure S3.** Time series of air mass regimes (a,l), input variables for the classification (b–i), and validation variables (j,k,m–v) from 1 to 13 March 2014. Unless labeled differently, the data was measured at the UFS except for $\Delta\theta_v$ that represents the maximum difference among the sites. The color shading highlights the most important air mass classes. Symbols are explained in Sect. 2.2 and 2.3 of the main article.